# Increasing sensitivity of dryland vegetation greenness to precipitation due to rising atmospheric $CO_2$

Yao Zhang [1,2,3] ✉, Pierre Gentine [4], Xiangzhong Luo [5], Xu Lian [1,4], Yanlan Liu [6], Sha Zhou [7], Anna M. Michalak[8], Wu Sun [8], Joshua B. Fisher[9], Shilong Piao [1,10] & Trevor F. Keenan [2,3] ✉

Water availability plays a critical role in shaping terrestrial ecosystems, particularly in low- and mid-latitude regions. The sensitivity of vegetation growth to precipitation strongly regulates global vegetation dynamics and their responses to drought, yet sensitivity changes in response to climate change remain poorly understood. Here we use long-term satellite observations combined with a dynamic statistical learning approach to examine changes in the sensitivity of vegetation greenness to precipitation over the past four decades. We observe a robust increase in precipitation sensitivity (0.624% $yr^{-1}$) for drylands, and a decrease (−0.618% $yr^{-1}$) for wet regions. Using model simulations, we show that the contrasting trends between dry and wet regions are caused by elevated atmospheric $CO_2$ (e$CO_2$). e$CO_2$ universally decreases the precipitation sensitivity by reducing leaf-level transpiration, particularly in wet regions. However, in drylands, this leaf-level transpiration reduction is overridden at the canopy scale by a large proportional increase in leaf area. The increased sensitivity for global drylands implies a potential decrease in ecosystem stability and greater impacts of droughts in these vulnerable ecosystems under continued global change.

Recent warming has led to large increases in atmospheric water demand and vegetation water consumption in various regions[1–3], potentially causing declines in surface soil moisture and increases in drought severity[4,5]. Vegetation plays a critical role in regulating water fluxes between soil and the atmosphere and is itself also directly affected by the amount of available water in the soil[6]. In low and mid-latitude regions, water is considered as the primary environmental factor that determines vegetation distribution, species composition and ecosystem functioning, even in tropical forests which is often not considered water-limited[7]. Large variations of vegetation greenness and radial growth of trees have been attributed to precipitation trends and anomalies[8–10]. The sensitivity

[1]Sino-French Institute for Earth System Science, College of Urban and Environmental Sciences, Peking University, Beijing, China. [2]Climate and Ecosystem Sciences Division, Lawrence Berkeley National Laboratory, Berkeley, CA, USA. [3]Department of Environmental Science, Policy and Management, UC Berkeley, Berkeley, CA, USA. [4]Department of Earth and Environmental Engineering, Columbia University, New York, NY, USA. [5]Department of Geography, National University of Singapore, Singapore, Singapore. [6]School of Earth Sciences, The Ohio State University, Columbus, OH, USA. [7]State Key Laboratory of Earth Surface Processes and Resources Ecology, Faculty of Geographical Science, Beijing Normal University, Beijing, China. [8]Department of Global Ecology, Carnegie Institution for Science, Stanford, CA, USA. [9]Schmid College of Science and Technology, Chapman University, Orange, CA, USA. [10]State Key Laboratory of Tibetan Plateau Earth System, Resources and Environment, Institute of Tibetan Plateau Research, Chinese Academy of Sciences, Beijing, China. ✉e-mail: zhangyao@pku.edu.cn; trevorkeenan@berkeley.edu

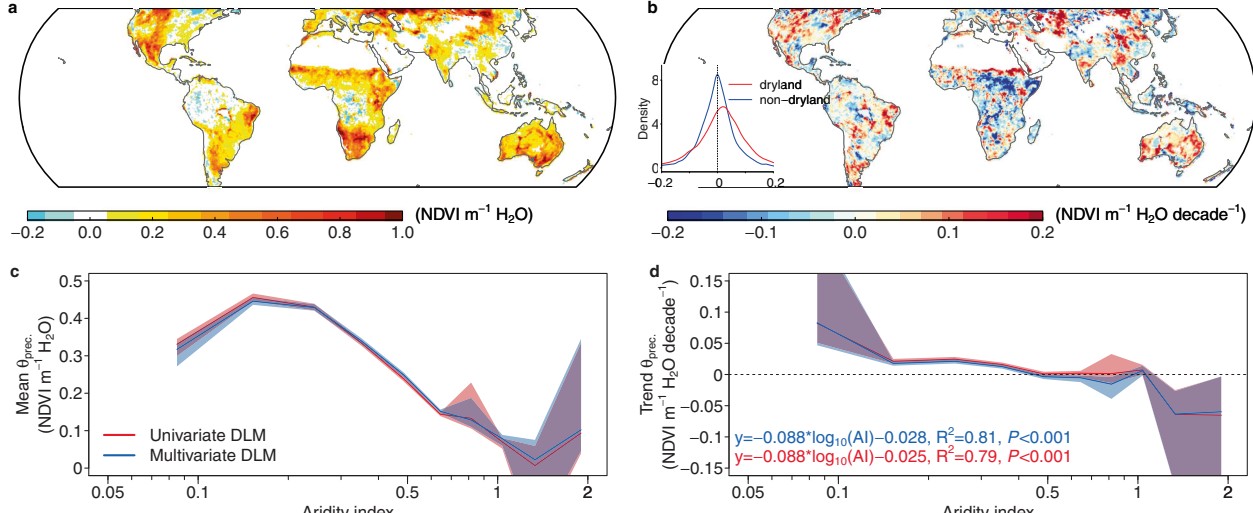

**Fig. 1 | Spatial patterns of precipitation sensitivity ($\theta_{prec}$).** Mean (**a**) and trend (**b**) of precipitation sensitivity during 1981–2015 obtained from a multivariate dynamic linear model (DLM). The inset in **b** shows the probability density of the precipitation sensitivity for dryland (red) and non-dryland (blue). The response of the (**c**) mean and (**d**) trend of precipitation sensitivity along the aridity index, where red indicates results from univariate model that only considers precipitation, and blue indicates results from the multivariate DLM. Shades indicate the 95% confidence interval calculated in each aridity bin through bootstrapping ($n = 5000$).

of vegetation greenness to precipitation is thus of critical importance, not only to understand ecosystem responses to drought and post-drought recovery, but also for predicting the dynamics of vegetation under a changing climate and the resulting impacts on the global carbon cycle[11,12].

Spatially, the sensitivity of vegetation productivity to precipitation variation has been reported to decrease as mean precipitation increases[13,14], though the relationship is influenced by other factors that regulate the ecosystem water availability, including environmental factors such as precipitation variability[15], mean temperature[16], soil texture[17] and ecosystem characteristics such as plant water use strategy[18] and rooting depth[19]. Temporally, in addition to the changes in hydroclimate and consequent plant acclimation[20], shifts in species composition[21], factors that lead to changes in plant water use efficiency and the coupling strength between vegetation and precipitation can also alter the sensitivity of vegetation greenness to precipitation. For example, nitrogen deposition and $CO_2$ fertilization[22], both of which increase leaf-level water use efficiency, also increase vegetation sensitivity to precipitation[13]. These effects, however, are further complicated by their interactions and the human-environment nexus[16,23]. The temporal changes in sensitivity of vegetation greenness to precipitation, as well as the underlying mechanism are still less known at a global scale. Considering the major role that vegetation plays in regulating the feedbacks between the atmosphere and the biosphere[24], this knowledge gap constitutes a large source of uncertainty in projection of future climate change.

Here we use a dynamic statistical learning approach, combined with remotely sensed normalized difference vegetation index (NDVI) and an ensemble of land-surface models, to quantify the sensitivity of vegetation canopy greenness to precipitation, and examine and attribute its temporal changes over 1981–2015. The approach is based on a dynamic linear model (DLM), which can estimate the time-varying relationship between environmental factors (e.g., precipitation, temperature, radiation) and NDVI remotely sensed from GIMMS (Methods), allowing us to derive a robust estimate of precipitation sensitivity. We then evaluate the trend of this precipitation sensitivity along a dryness gradient and use a parsimonious ecohydrological model to understand what drives its changes over time. Our results show that the precipitation sensitivity has continuously increased for the drylands and decreased for the wet regions during the past four decades, and eCO$_2$ is the major driving factor to explain these contrasting trends.

## Results

### Temporal changes of vegetation sensitivity to precipitation

The mean sensitivity of vegetation canopy greenness to precipitation ($\theta_{prec}$) calculated from the DLM using satellite observed NDVI is found to be the highest in arid regions (Fig. 1a), and this sensitivity decreases for regions that are more humid, as indicated by a higher aridity index (Fig. 1c). Aridity index is an indicator of the degree of dryness calculated as the ratio between long-term precipitation and potential evapotranspiration[25]. Here we use a static map of aridity index provided by CGAIR (Supplementary Fig. 1). Previous studies using field observations also found decreased rain use efficiency with increasing wetness, corroborating our remote sensing-based analysis[13].

Importantly, we find that $\theta_{prec}$ increases over time in many drylands (aridity index < 0.65, Supplementary Fig. 1), e.g., western US, Australia, central Asia and northwest China (Fig. 1b). Negative trends are mostly found in non-drylands (aridity index > 0.65), e.g., pan tropics or southern China. Exceptions exist in regions of Africa (e.g., the Sahel, Horn of Africa), where strong decreases happen in relatively dry regions. This is likely due to the significant increase of precipitation in these regions after 1980[26] (Supplementary Fig. 2). Collectively, the trend in $\theta_{prec}$ shifts from positive to negative across a wetness gradient (Fig. 1d), suggesting that vegetation greenness in drylands becomes overall more sensitive to precipitation variations, while greenness in non-drylands becomes less sensitive. This pattern is consistent when $\theta_{prec}$ is estimated using either a univariate DLM (including only the autocorrelation and precipitation terms) or a multivariate DLM (additional terms including cloud fraction and temperature) (Fig. 1). These results are robust regardless of the vegetation and precipitation datasets, or the sensitivity estimating method being used (Supplementary Discussion, Supplementary Figs. 3–11).

The increase of $\theta_{prec}$ in drylands suggests a potential increase of vegetation greenness variability. To test this, we analyze the trends in NDVI variability caused by precipitation variations ($\sigma NDVI_{prec}$) using results from the DLM (Fig. 2a). The spatial pattern of $\sigma NDVI_{prec}$ trends are more similar to the $\theta_{prec}$ trend than the precipitation variability trend ($r = 0.42$ vs 0.10, $P < 0.001$, Supplementary Fig. 12), indicating

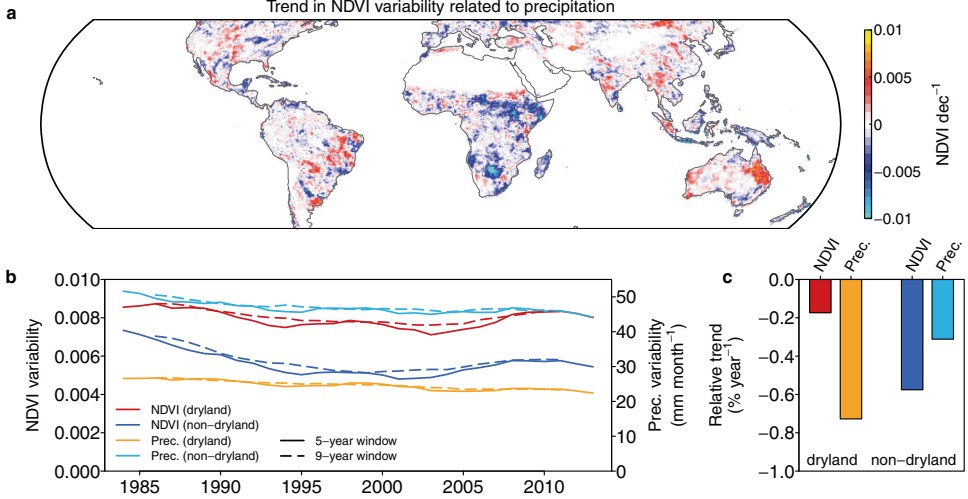

**Fig. 2 | Change in vegetation variability caused by precipitation. a** Trend in NDVI variability caused by precipitation ($\sigma NDVI_{prec}$). $\sigma NDVI_{prec}$ is calculated as the standard deviation of the NDVI anomalies contributed by precipitation ($\theta_{prec}$ × precipitation anomaly) within each 60-month window. **b** Time series of $\sigma NDVI_{prec}$

and precipitation variability for drylands and non-drylands. Dashed and solid lines indicate time series estimated from a moving window of 108 and 60 months (9 and 5 years), respectively. **c** Normalized trend in $\sigma NDVI_{prec}$ (NDVI) and precipitation variability (Prec.) for dryland and non-dryland.

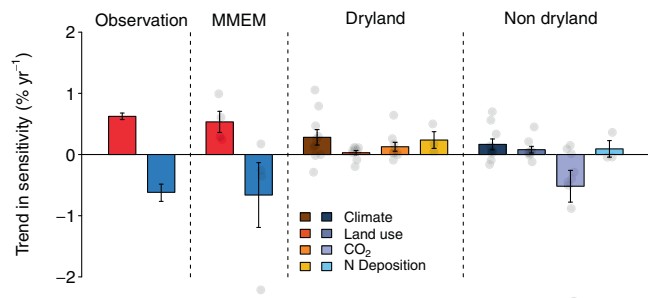

**Fig. 3 | Contrasting trends in precipitation sensitivity for drylands and non-drylands.** Comparison between the observed and modeled precipitation sensitivity trend as a percentage of the multi-year average. For observations and the multi-model ensemble mean (MMEM), red and blue indicate dryland and non-dryland regions, respectively. MMEM and attribution are obtained from MsTMIP model scenario simulations. For observations, bars represent the median trends and error bars indicate the 95% confidence interval of spatial variation through bootstrapping. For models, the bars and error bars indicate the mean and standard error of the mean (SEM) of median trend across models, respectively. Semi-transparent dots represent the median trend from each model. Since not all models provide simulations for all four scenarios, to delineate the contribution for each driving factor, different models may be used (Methods). The sample sizes for each factor are: Observation 17601 and 14095 (for dryland and non-dryland, respectively); MMEM 4 and 4; Climate 10 and 10; Land use 9 and 9; $CO_2$ 9 and 9; N Deposition 3 and 3.

that changes in $\theta_{prec}$ drive the different trend in NDVI variability between the drylands and non-drylands (unpaired one-sided t-test, $P < 0.001$). We also compare the trajectories of $\sigma NDVI_{prec}$ changes through time, together with the changes in precipitation variability (Fig. 2b). $\sigma NDVI_{prec}$ shows a decrease for both drylands and non-drylands, with a greater decline for the latter. This decreasing trend for both regions is caused by a decrease in precipitation variability during the same period. By comparing the normalized trend of $\sigma NDVI_{prec}$ and precipitation variability for both dryland and non-dryland ecosystems, we find that for drylands, precipitation variability reduces at a rate of −0.73% yr⁻¹ as compared to a much smaller reduction for $\sigma NDVI_{prec}$ (−0.17% yr⁻¹) (Fig. 2c). This suggests that the increase in $\theta_{prec}$ partially offsets the impact of reduction in precipitation variability on $\sigma NDVI_{prec}$.

## $CO_2$ as the major contributor for the contrasting trends in precipitation sensitivity

A number of factors can cause changes in precipitation sensitivity ($\theta_{prec}$). For example, the $\theta_{prec}$ trend shows a strong dependence on the trend of water availability. Regions with decreasing annual precipitation and root-zone soil moisture tend to show positive $\theta_{prec}$ trend[27] (Supplementary Fig. 13a, c). This can be explained by an increase of rain use efficiency, which amplifies vegetation responses to precipitation, during drought years[13]. However, because precipitation trends are not significant in both drylands and non-drylands, precipitation change alone cannot explain the contrasting trends in $\theta_{prec}$ (Supplementary Fig. 2c). Changes in potential evapotranspiration and its components (e.g., radiation, temperature) can also affect the water availability, but these factors show very weak relationships with $\theta_{prec}$ in terms of their trends (Supplementary Fig. 14). Additionally, we find that the $\theta_{prec}$ trend within each biome shows a similar pattern along the aridity index as that from the entire study regions (Supplementary Fig. 15), suggesting that biome-specific characteristics are not the major cause for the contrasting trends.

To better understand the cause of the divergent changes in $\theta_{prec}$ between drylands and non-drylands, we use model-simulated leaf area index (LAI) from four scenarios in the Multi-scale Synthesis and Terrestrial Model Intercomparison Project (MsTMIP)[28] and calculate $\theta_{prec}$ in a similar manner (Methods). The simulated LAI is used in this causal attribution as it is comparable to NDVI − both represent canopy structural changes. The multi-model ensemble mean (MMEM) of $\theta_{prec}$ estimated from the historical forcings exhibits large uncertainty but generally reproduces the contrasting $\theta_{prec}$ trends, i.e., positive in drylands and negative in non-drylands, with similar magnitudes comparable to those derived from remote sensing observations (Fig. 3). Using a combination of factorial simulation scenarios (Supplementary Table 1), we analyze the contribution of four factors (climate change, elevated atmospheric $CO_2$, land use change and nitrogen deposition) to the $\theta_{prec}$ trend, for drylands and non-drylands separately.

We find that changes in climate forcing contribute positively to $\theta_{prec}$ for both drylands and non-drylands (Fig. 3), possibly due to global warming and the resulting drying trend of the atmosphere[2]. Given $\theta_{prec}$ increases with climate dryness (Fig. 1c), such a drying trend likely leads to increasing $\theta_{prec}$ globally. Nitrogen deposition, though only represented in four out of ten models, also shows a positive contribution to $\theta_{prec}$, especially for drylands. However, as nitrogen dynamics are still

poorly represented in these models, such estimate should be considered to be relatively uncertain. Land use change plays a relatively small role on $\theta_{prec}$ changes in both drylands and non-drylands. The largest difference between drylands and non-drylands, however, comes from the impact of elevated atmospheric $CO_2$, which increases $\theta_{prec}$ in drylands, but decreases $\theta_{prec}$ in non-drylands. We also test the robustness of this attribution method by using only the models that have simulations for all four scenarios. The sign of the contributions remains unchanged for the selected ensemble, while the magnitudes vary (Supplementary Fig. 16).

Given the large effect of $CO_2$ in explaining the difference between drylands and non-drylands, we further quantify its impact on disentangled components controlling $\theta_{prec}$ by levering the model outputs. Here, $\theta_{prec}$, approximated by LAI sensitivity to precipitation, can be further decomposed to three components with well-defined ecohydrological meaning (the inverse of transpiration per leaf area, transpiration over evapotranspiration, and evapotranspiration ratio in their partial derivative form):

$$\theta_{prec} \approx \frac{\partial LAI}{\partial P} = \frac{\partial LAI}{\partial E_T} \frac{\partial E_T}{\partial E} \frac{\partial E}{\partial P} \qquad (1)$$

The results show that elevated $CO_2$ in the atmosphere (e$CO_2$) increases the sensitivity of LAI to transpiration ($\frac{\partial LAI}{\partial E_T}$) for both drylands and non-drylands, while the effect on transpiration ($E_T$) sensitivity to evapotranspiration ($\frac{\partial E_T}{\partial E}$) is positive for drylands and negative for non-drylands. The largest difference comes from evapotranspiration ($E$) sensitivity to precipitation ($\frac{\partial E}{\partial P}$), which explains most of the contrasting trends for drylands and non-drylands (Supplementary Fig. 17).

## Mechanistic explanation for the $CO_2$ effect on precipitation sensitivity

For a mechanistic understanding of how e$CO_2$ leads to contrasting trends of vegetation sensitivity in drylands and non-dryland regions, we use a minimalistic hydrological model[29] and evaluate the change of precipitation sensitivity under different $CO_2$ levels. This model is based on a water balance equation and is designed to analytically derive the relationship between evapotranspiration and precipitation, similar to the Budyko curve. The modified version by Good et al.[30] can further partition evapotranspiration into interception, transpiration and soil evaporation based on the characteristics of climate, soil and vegetation. We improve the model so that it can account for the $CO_2$ effect on stomatal conductance and on LAI (Methods). With different combinations of the parameter settings, including rooting depths, soil types and rainfall depths, the model is used to calculate the partial derivative of LAI over precipitation as a proxy of $\theta_{prec}$ (see Methods).

The predicted LAI sensitivity to precipitation exhibits a maximum in semiarid regions, consistent with a maximum fraction of biological water use to precipitation in mesic environments[30] (Fig. 4a). This sensitivity pattern is also similar to what we derived using satellite observed LAI (Supplementary Fig. 6). Using this simple model, we can decompose the sensitivity changes into three different $CO_2$ effects, and understand what mechanism drives the contrasting trends between dryland and non-dryland. The first effect is that e$CO_2$ can reduce the aperture of the stomata (reduction in stomatal conductance), so that the amount of water needed by plants decreases, and thus so does the sensitivity of transpiration to precipitation ($\frac{\partial E_T}{\partial P}$) (Fig. 4b). The second effect is that e$CO_2$ can stimulates vegetation cover, the relative increase of LAI is suggested to be strongest in drylands and diminishes quickly as it gets wetter[31]. This effect overrides the decline of $\frac{\partial E_T}{\partial P}$ in dry regions, creating a contrasting trend for drylands and non-drylands. The third effect is also due to the e$CO_2$-induced stomatal conductance reduction, which on the other hand, reduces transpiration per leaf area. This is equivalent to a universal increase of water use efficiency and $\frac{\partial LAI}{\partial E_T}$. The combination of these

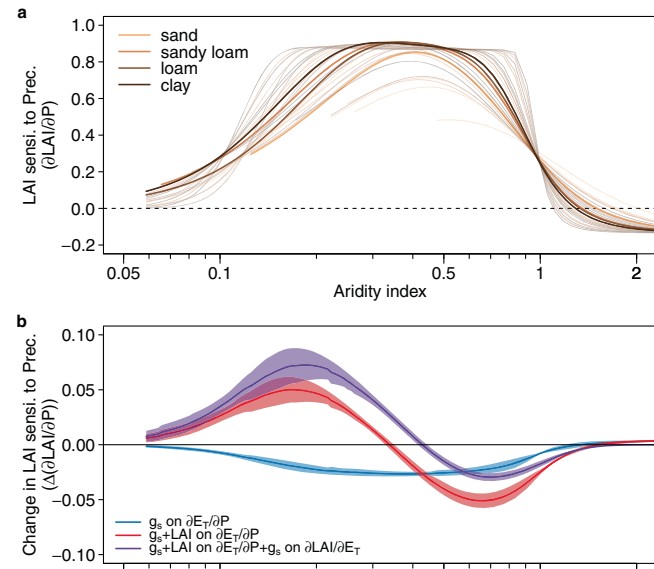

**Fig. 4 | Predicted responses of vegetation sensitivity by the minimalistic hydrological model. a** predicted responses along the aridity index. Different line colors represent different soil types, with thicker lines indicating the estimates based on global land mean values for rooting depth and rainfall depth. Thin lines indicate estimates using other possible value combinations (Supplementary Table 3). **b** predicted changes of sensitivity due to elevated atmospheric $CO_2$. Three types of $CO_2$ effect are considered, with each line represents a stepwise combination of them (see Methods). The solid lines with shaded areas indicate the ensemble mean and standard deviation of all soil and climate combinations. The results are obtained assuming all vegetation are C3.

effects can broadly capture the contrasting trend of $\theta_{prec}$ for drylands and non-drylands. Although previous studies suggest that $CO_2$ may also decrease potential evapotranspiration and change aridity[32,33], we find its contribution is one order of magnitude smaller than the direct and indirect $CO_2$ effect on stomata and leaf area (5.6% and 13.4% for drylands and non-drylands, respectively, Supplementary Fig. 18). We note that taking the differences between C3 or C4 photosynthesis pathways or different levels of interception fractions into consideration may alter the contribution of each component, but the general patterns remain unchanged (Supplementary Fig. 18). It is worth noting that this simple model does not account for the variations in plant hydraulic traits and inter-species competition for water, which may both affect the vegetation sensitivity to precipitation[18,34], but their effects on the trend of sensitivity are still uncertain.

## Discussion

The e$CO_2$ has both direct and indirect effects on plant-water relationships. The direct effect increases the partial pressure of intercellular $CO_2$ and water use efficiency[35], which can lead to increases of leaf-level photosynthesis and decreases of transpiration[36]. However, these leaf-level responses may be different at the ecosystem level, especially when the competition between individuals is taken into account[34]. For example, water saved by one plant may be used by another. The effect of competition on individual plant growth may depend on the hydraulic diversity, species characteristics and local environment[37,38]. Invariably however, increased leaf-level water use efficiency leads to increases of the ecosystem LAI, especially in water-limited regions, and increases in available soil moisture in energy-limited regions. Both are often considered as indirect effects of $CO_2$[25,39–41]. Previous modeling studies suggest that indirect effects play a much more important role in arid regions than in humid regions[42]. These effects can also be amplified due to the difference between C3

and C4 plants, whose distribution follows the gradient of aridity in low latitude regions. C4 plants benefit much less from the direct eCO$_2$-induced photosynthesis increase, but show a stronger decline in stomatal conductance and thus indirect water savings, allowing greater increases of vegetation cover in the drylands[43,44]. The strong increases in leaf area, in turn, lead to an increase in water demand and water loss through cuticular conductance and stomatal leakiness[45], offsetting the CO$_2$ water saving effect due to reduced stomatal aperture[31,42]. Increased vegetation covers in drylands can also lead to increased infiltration and reduced runoff[46,47], creating a positive feedback, further enhancing the vegetation sensitivity to precipitation.

On the other hand, in wet regions, although photosynthesis increases under eCO$_2$ due to the direct effect, this does not necessarily lead to a proportional increase in leaf area and canopy conductance considering the high LAI baseline[48]. In these regions, leaf biomass increase is limited by light and nutrient availability, sink strength (i.e., the capacity of enzymes to assimilate carbon), as well as the capability of plants to use and allocate excess carbon to other organs[43,49]. The direct water saving effect together with a non-significant increase in leaf area, reduces the total water consumption[42], and leads to increased runoff globally[50,51]. Excess water further decouples transpiration and precipitation, and therefore decreases the vegetation sensitivity to precipitation and increases the safety margin to climate variations. The diverging responses of vegetation sensitivity along dryness gradients are caused by the different strength of the indirect water saving effect, in particular whether it can stimulate leaf area increases. The minimalistic model we employ offers a similar explanation from a hydrological perspective: water savings in dry regions allow for a higher vegetation cover and a greater fraction of precipitation to be transpired by vegetation, while for wetter regions where ecosystems are mostly energy limited, water savings essentially decrease the transpiration fraction and hence the sensitivity of transpiration to precipitation. This contrasting response is also supported by the observed divergent trends in global runoff for drylands and non-drylands[52,53]. Other factors including soil texture, rooting depth, water table depth, precipitation seasonality, and stomatal sensitivity to drought also affect ecosystem water availability and precipitation sensitivity, and their interactions with CO$_2$ needs to be further explored. It should also be noted that although trends in precipitation and potential evapotranspiration do not likely explain this contrasting trend of $\theta_{prec}$ at a global scale, they may play an important role in regulating the local water availability and thus contribute to the variation of $\theta_{prec}$ at local scale.

Our analysis based on satellite vegetation greenness observations should not be interpreted as long-term changes in the sensitivity of photosynthesis to precipitation. This is primarily due to the fact that vegetation greenness as indicated by NDVI does not reflect the direct CO$_2$ physiological effect that stimulates plant photosynthesis[54], which plays a critical role in assessing the long-term trend[55]. The strong CO$_2$ induced-increase in light use efficiency may override the declining trend of precipitation sensitivity in wet regions. This is also supported by our MsTMIP analysis using gross primary production (GPP) instead of LAI (Supplementary Fig. 19). GPP sensitivity to precipitation increases for both dryland and non-dryland regions, and the eCO$_2$-induced negative trend in non-dryland regions also shifts to positive values when we use GPP instead of LAI, although its magnitude is still smaller than the drylands.

The large increase in sensitivity of vegetation greenness to precipitation for drylands indicates that under rising CO$_2$, the same amount of precipitation would translate to an increased vegetation greenness. This will lead to greater biomass, increased infiltration and soil moisture[47], and greater carbon uptake. On the other hand, due to the increased vegetation sensitivity to precipitation, precipitation variability would also lead to greater variability of vegetation greenness in drylands. For example, stronger drought impacts can be

expected when precipitation is anomalously low. This effect can be exacerbated by the excessive growth of vegetation in the previous period, which depletes soil moisture and increases drought stress, a phenomenon known as structural overshoot[56,57]. A greater variability in global agricultural productivity may also be anticipated considering the large area of rained croplands and pastures in drylands. As semi-arid regions have a large contribution to interannual variability in the global carbon sink[9,58], the increase in precipitation sensitivity, together with more frequent and more severe climate extremes, could further amplify the relative importance of drylands in the terrestrial carbon cycle. A "greening but drying" trend may thus be more prevalent in drylands in the future[25,41], potentially increasing drought risk.

## Methods
### Datasets
We used GIMMS 3 g v1 normalized difference vegetation index (NDVI) as an indicator of vegetation activity[59]. This dataset uses reflectance measurement from a series of Advanced Very High Resolution Radiometer (AVHRR) sensor onboard the NOAA polar orbiting satellites spanning from 1981 to 2015. We used this datasets as the primary data since previous studies have found that GIMMS3g showed best temporal consistency in comparison with other long-term NDVI dataset such as LTDR4 (long-term data record version 4), VIP3 (vegetation index and phenology version 3)[60]. Although drifts in satellite orbits may change the sun-sensor geometry and affect the surface reflectance, this effect is minimized through the normalization of NDVI calculation. We also filtered out NDVI values when air temperature is below zero to eliminate potential snow effects. It should be noted that although NDVI is often used as a proxy of vegetation photosynthetic capacity, it does not reflect the direct CO$_2$ fertilization effect through increases in carboxylation rate[54], and therefore cannot be directly compared with the response of gross primary production (GPP) to CO$_2$. This does not mean, however, that we cannot use NDVI to understand the CO$_2$ effect on vegetation, since the increases in carbon fixation will ultimately be used and allocated to biomass. This effect is considered as the indirect effect of CO$_2$ and has been suggested to be the major contributor to the increase of global LAI[61].

In addition to this NDVI dataset, we also used a long-term GIMMS LAI 3g dataset (1982–2015). This dataset is developed based on a machine learning algorithm that links GIMMS NDVI 3g dataset to MODIS LAI during the period of 2000–2009. Although the resulting GIMMS LAI 3g dataset strongly depends on the performance of GIMMS NDVI 3g and MODIS LAI, it compared well against field observations and partially alleviates the saturation effect of NDVI in densely vegetated regions[62]. Both datasets were aggregated to 0.5° × 0.5° spatial resolution and monthly temporal resolution to match the climate datasets.

We used climate variables from CRU TS 4.04 to calculate the precipitation sensitivity[63]. The CRU dataset is generated from weather stations using spatial interpolation methods. Since precipitation is crucial to the calculation of vegetation sensitivity to precipitation, we also tested another observational-based dataset from the Global Precipitation Climatology Centre (GPCC)[64], which uses a larger number of weather stations. The results show similar spatial patterns, further supporting the robustness of our analysis (Supplementary Fig. 8).

Aridity index reflects the ecosystem hydrological balance between water supply (precipitation, $P$) and demand (potential evapotranspiration, $E_P$). We follow the conventional definition of aridity index ($P/E_P$) and use the aridity index dataset provided by CGIAR Consortium for Spatial Information (Supplementary Fig. 1). It should be noted that aridity index is a measure of long-term climatic dryness conditions and different potential evapotranspiration definitions and calculations can yield different potential evapotranspiration trends, we ignored aridity changes during the study period for simplicity.

## Precipitation sensitivity of vegetation

The vegetation sensitivity to precipitation investigated in this study is a proxy of the response of vegetation greenness to a perturbation in precipitation. Since the response to the total and to perturbation may be different, we estimated this sensitivity using de-seasonalized and detrended anomalies ($\delta$) of NDVI and climate variables. Taking the legacy effect from the previous month into consideration, the anomaly of current month NDVI ($\delta NDVI_t$) can be expressed using a lag-1 autocorrelation model:

$$\delta NDVI_t = \theta_{NDVI}\delta NDVI_{t-1} + \theta_{prec}\delta Prec_{t-1} + \theta_{temp}\delta Temp_t + \theta_{cloud}\delta Cloud_t + \varepsilon \tag{2}$$

where $\delta NDVI_t$, $\delta Prec_{t-1}$, $\delta Temp_t$ and $\delta Cloud_t$ represent anomalies for NDVI, precipitation, temperature, and cloud fraction, respectively, with subscripts indicate whether the anomaly is calculated from the current month ($t$) or the previous month ($t-1$). $\varepsilon$ is the error term for the model. The coefficient for $\delta Prec_{t-1}$ ($\theta_{prec}$) is the precipitation sensitivity on which our analysis is based. Since precipitation from previous months may also contribute to the water supply and affect the vegetation growth, we also considered five other lengths to calculate precipitation anomalies (current month, current and previous 1 month, current and previous 2 months, current and previous 3 months, previous 1–2 months). We found that for most dry regions, models using the previous month's precipitation anomaly have the best performance (Supplementary Fig. 20). We therefore used the previous month in the dynamic linear model analysis (see below). For the multivariate linear regression analysis, precipitation anomalies from all lengths were used and for each pixel, $\theta_{prec}$ from the best model is selected.

We also built another model that only considers previous month's precipitation (univariate model) and autocorrelation to test the robustness of our results:

$$\delta NDVI_t = \theta_{NDVI}\delta NDVI_{t-1} + \theta_{prec}\delta Prec_{t-1} + \varepsilon \tag{3}$$

The sensitivity obtained from this univariate model is similar to that from the multivariate model (Fig. 1 and Supplementary Fig. 3).

We used both dynamic linear model (DLM) and multivariate linear regression to calculate the sensitivity. DLM is a statistical approach to model time-series signals by considering various contributing factors[65,66]. It was originally developed for solving economic problems and recently introduced to the Earth and environmental sciences[57,67]. For each pixel, we built a DLM which considers the trend, seasonal variation of NDVI, and contributions from climate variables and lag-1 autocorrelation factors to predict the de-seasonalized detrended anomalies. The DLM is based on the Bayesian Theorem and predicts a time-vary relationship at each timestep (one month) between the input variables and target variable. This allows us to characterize the temporal changes of vegetation sensitivity to precipitation, and precipitation contribution to the NDVI anomalies (sensitivity × precipitation anomalies). In addition to the DLM, we also used a multivariate regression model to calculate the precipitation sensitivity and its trend using Eqs. (2) and (3). The detailed description of DLM and multivariate regression method can be found in Supplementary Text 1 and 2. The DLM modeling was carried out under Python 3.7 environment.

## Terrestrial biosphere model analysis

We used a suite of terrestrial biosphere models from the MsTMIP model intercomparison project to understand the causes of the trend in precipitation sensitivity of vegetation. The models that participated in MsTMIP conduct simulations under different scenarios, which can be used to decompose the observed trend in vegetation precipitation sensitivity into different driving factors. In practice, we used a combination of four scenario simulations from MsTMIP (Supplementary Table 1), with results from SG1 representing sensitivity changes due to climate factors; differences in SG2-SG1, SG3-SG2, and BG1-SG3 representing sensitivity changes due to land use change, $CO_2$, and nitrogen deposition, respectively. For each model-scenario combination, we used leaf area index (LAI) simulations together with the temperature, precipitation and downward shortwave radiation data from CRU-NCEP V6. CRU-NCEP V6 data was used here instead of the CRU TS dataset because it is the climate forcing for MsTMIP. The precipitation sensitivity was calculated using multivariate linear regression at monthly scale for each decade during 1980–2010, from which we calculated the mean and trend. To make the modeling outputs comparable with the results we obtained from NDVI, the trend of $\theta_{prec}$ calculated from LAI is normalized by the mean $\theta_{prec}$ to get a relative trend for each pixel. We then calculated the median value for drylands and non-dryland regions separately, for each driving factor using the scenario combinations mentioned above. For models that do not provide simulations for specific scenarios, we only calculated those that are available (Supplementary Table 2).

To further understand the $CO_2$ effect on $\theta_{prec}$, we used additional model outputs, including transpiration ("Veg", $E_T$), evapotranspiration ("Evap", $E$) and calculated $\frac{\partial LAI}{\partial E_T}, \frac{\partial E_T}{\partial E}, \frac{\partial E}{\partial P}$ for the SG3 and SG2 scenario for each model. The differences between these two scenarios can be used to understand the responses of these $\theta_{prec}$ components to $CO_2$. Only eight models that have both SG3 and SG2 simulation scenarios and predictions of $LAI$, $E_T$, and $E$ were used for this analysis (Supplementary Fig. 14).

## Analytical derivation of vegetation sensitivity to precipitation

We conducted a theoretical analysis to demonstrate how $CO_2$ affects the vegetation sensitivity to precipitation ($\theta_{prec}$) differently in dryland and non-dryland. This is based on a minimalistic hydrological model which represents simplified ecohydrological processes so that the relationship between vegetation and precipitation can be analytically derived. In this study, we modified the model and incorporated the direct and indirect $CO_2$ effects. This allowed us to understand how $CO_2$ affects drylands and non-dryland differently. The vegetation sensitivity to precipitation ($\theta_{prec}$) can be calculated as:

$$\theta_{prec} = \frac{\partial NDVI}{\partial P} \approx \frac{\partial LAI}{\partial E_T}\frac{\partial E_T}{\partial P} \tag{4}$$

where $LAI$, $E_T$ and $P$ indicate the vegetation leaf area index, canopy transpiration and precipitation, respectively. $LAI$ and $E_T$ are linked through average transpiration per leaf area ($E_L$):

$$E_T = LAI \cdot E_L \tag{5}$$

The partial derivative of $LAI$ to $E_T$ can be simplified to:

$$\frac{\partial LAI}{\partial E_T} = \frac{1}{E_L} \tag{6}$$

$E_L$ is expected to have limited variations across aridity gradient but would show a universal decrease due to the $CO_2$ effect on stomatal closure over time. This $CO_2$ effect is quantified analytically below.

To understand the response of $\frac{\partial E_T}{\partial P}$, we used the minimalistic ecohydrological model originally developed by Porporato et al.[29], and recently modified by Good et al.[30]. This modification further separates the contribution of transpiration ($E_T$), evaporation ($E_S$) and interception ($E_I$) to evapotranspiration, which is essential to understand the response of $E_T$ sensitivity to $P$. This model is based on a well-developed ecohydrological soil water balance theory which calculates soil water dynamic with precipitation water input and water losses from leakage,

interception, transpiration and soil evaporation. Soil moisture dynamics and ecosystem aridity determines the partitioning of precipitation to these components. Specifically, fraction of plants' transpiration to precipitation can be calculated as:

$$\frac{E_T}{P} = f\bar{x}\phi'\rho \tag{7}$$

where $f$ is the fraction of transpiration ($E_T$) to evaporation and transpiration ($E_S + E_T$), which depends on the soil moisture. $\bar{x}$ is the average effective soil moisture, $\phi'$ is the aridity index adjusted for interception and $\rho$ is the throughfall fraction. The latter three controls the fraction of $E_S + E_T$ to precipitation. The probability distribution of soil moisture, which determines $f$ and $\bar{x}$, is calculated based on precipitation frequency, intensity, and soil traits (see detailed derivation in Supplementary Method 3).

The original model does not consider effect of $CO_2$ on transpiration, here we modified the model by taking $CO_2$ effect into account. $CO_2$ affects $E_T$ through both direct effect of reducing stomatal conductance ($g_s$) and indirect effect of increasing LAI. To consider these two effects in the minimalistic model, we used a scaling factor $\kappa$ to represent the fraction of stomatal closure due to $CO_2$ and a scaling factor $\zeta$ to represent the proportional LAI increase due to $CO_2$. Factor $\kappa$ can be calculated using a stomatal conductance model[68] and factor $\zeta$ can be expressed as a function of aridity index based on a synthesis of multiple FACE experiments[31] (see details in Supplementary Method 3). $\kappa$ is also used to estimate $CO_2$ effect on $E_L$. The modified minimalistic model is expressed as:

$$\frac{E_T}{P} = f\bar{x}\phi'\rho\kappa\zeta \tag{8}$$

Since each component on the right-hand-side of Eq. (8) is a constant or can be expressed as a function of aridity index, we can analytically derive the partial derivative of transpiration to precipitation ($\frac{\partial E_T}{\partial P}$), which is required to calculate the vegetation sensitivity to precipitation ($\theta_{prec}$) using Eq. (4).

We also considered the effect of $CO_2$ on $E_P$ and consequently on $\phi$. We predicted $E_P$ using a revised Penman–Monteith model[33], in which $CO_2$ is expected to reduce canopy conductance and affect potential evapotranspiration.

$$E_P = \frac{0.408\triangle R_n^* + \gamma_{pc}\frac{900}{T+273}uD}{\triangle + \gamma_{pc}\{1 + u[0.34 + 2.4 \times 10^{-4}([CO_2] - 300)]\}} \tag{9}$$

where $\triangle$ is the rate of change of saturation vapor pressure with temperature (Pa K$^{-1}$), $R_n^*$ is the net radiation adjust for ground heat flux (W m$^{-2}$), $\gamma_{pc}$ is the psychrometric constant ($\approx$66 Pa K$^{-1}$), $u$ is the wind speed at 2 m (m s$^{-1}$). $D$ is the vapor pressure deficit of the air (Pa). The term $2.4 \times 10^{-4}([CO_2] - 300)$ is used to account for the effect of $CO_2$ on surface resistance. We used fixed climate variable values and calculated the $E_P$ under $CO_2$ at 354 ppm and 384 ppm. The changes of $E_P$ directly affect the dryness and further changes the vegetation sensitivity to precipitation.

To understand the $CO_2$ effect on the precipitation sensitivity through different pathways, we evaluated the change of precipitation sensitivity between high $CO_2$ and low $CO_2$, with four $CO_2$ effect taken into account, i.e., (1) direct $CO_2$ effect on $\frac{\partial E_T}{\partial P}$ through change of $g_s$; (2) indirect $CO_2$ effect on $\frac{\partial E_T}{\partial P}$ through change of LAI; (3) direct $CO_2$ effect on $\frac{\partial LAI}{\partial E_T}$ through change of $g_s$; and (4) indirect effect of $CO_2$ on PET through change of ecosystem conductance. It should be noted that the fourth effect should have been a result of the first effect and here we only evaluated its magnitude for comparison. The numerical simulation was conducted under R 3.5.2.

## Reporting summary

Further information on research design is available in the Nature Research Reporting Summary linked to this article.

## Data availability

The GIMMS NDVI 3g v1 dataset is available at http://poles.tpdc.ac.cn/en/data/9775f2b4-7370-4e5e-a537-3482c9a83d88/, the CRU climate dataset is available at https://crudata.uea.ac.uk/cru/data/hrg/, the GPCC precipitation data is available at https://www.dwd.de/EN/ourservices/gpcc/gpcc.html, the MsTMIP model outputs are available at https://doi.org/10.3334/ORNLDAAC/1225, the CRU-NCEP V6 dataset is available through https://doi.org/10.3334/ORNLDAAC/1220, The aridity index data is available at https://doi.org/10.6084/m9.figshare.7504448.v3.

## Code availability

The codes for the analyses are available at GitHub (https://github.com/zhangyaonju/prec_sensitivity/) and have been archived on Zenodo (https://doi.org/10.5281/zenodo.6936321).

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

## Acknowledgements

This work is supported by High-performance Computing Platform of Peking University. Y.Z., T.F.K., W.S., and J.B.F. acknowledge support from the NASA IDS Award NNH17AE861. Y.Z. acknowledges additional support from the Special Funds of National Natural Science Foundation China (42141005). T.F.K. acknowledges additional support from the US Department of Energy (DOE) under Contract DE-AC02-05CH11231 as part of the RUBISCO SFA, a DOE Early Career Research Program award #DE-SC0021023, and an NSF PREEVENTS award #1854945. P.G. acknowledges support from NASA ROSES #17-THP17-0036. Funding for the Multi-scale synthesis and Terrestrial Model Intercomparison Project (MsTMIP; https://nacp.ornl.gov) activity was provided through NASA ROSES Grant no. NNX10AG01A. Data management support for preparing, documenting and distributing model driver and output data was performed by the Modeling and Synthesis Thematic Data Center at Oak Ridge National Laboratory (ORNL; http://nacp.ornl.gov), with funding through NASA ROSES Grant no. NNH10AN681. Finalized MsTMIP data products are archived at the ORNL DAAC (http://daac.ornl.gov). We thank Dr. Chi Chen for providing the GIMMS LAI 3g dataset.

## Author contributions

Y.Z. designed the study, performed the analysis and wrote the manuscript. Y.Z., T.F.K. and P.G., participated in the early-stage discussion. Y.L. contributed to the analysis tool. All co-authors commented on the results and contributed to the writing of the manuscript.

## Competing interests

The authors declare no competing interests.
