## [Peer Review File · Nature Communications]

Reviewer comments, first round -

Reviewer #1 (Remarks to the Author):

The authors of the submitted study conducted a series of statistical and theory driven analyses of vegetation patterns (1981-2015) and their response to vegetation. Dynamic and static, linear and non-linear empirical models are used to show that over the study period vegetation (as measured by satellite NDVI) is more sensitive to precipitation in drylands, with this sensitivity increasing more in drylands. These are complemented with analyses of a suite of terrestrial biosphere models as well as an update to a minimalistic model. These three approaches are all fairly consistent with each other. The authors then evaluated the drivers to the increased trend in sensitivity in drylands in the biosphere and minimalistic models and concluded that these changes are due to the role of CO₂ on vegetation.

It is very clear the authors have done a huge amount of work investigating the stated problem. The paper is fairly well written, however the overabundance of methods and approaches employed somewhat obscures the findings. For instance, the univariate and multivariate methods produce similar results (L102) as do the MLR and DLR produce similar results (L88). This suggests that similar conclusions can be drawn with simple methods. Additionally, the comparison between the different models in F3 mixes and matches and their presentation and isn't consistent (F13 is much more-clear in this regard). The section on the minimalistic modeling is interesting (and could be its own theoretical paper for instance) and should be made more descriptive by removing the partial derivative terms from the text where possible and instead reframing these processes by descriptive names. I suggest the authors clean up and simplify the methods presented (perhaps moving more to the supplementary) in order to create more space for improve descriptions and discussions. In fact much of the first part of the results (the finding that water limited ecosystems are more sensitive to water limitations) could be compressed to focus on the drivers of this.

In particular, the finding that when water isn't a limiting factor then ecosystems are not sensitive to water is not particularly new or novel. This is expected based on a simple Budyko analysis from which the authors incorrectly draw their definition of drylands (aridity is PET/P not P/PET). Water limited regions are sensitive to water and light limited regions are sensitive to light, so as they move from light limited to water limited under climate change their sensitivities go up, with stronger increases for more arid drylands (as shown in S12). In the results (and in F2) it is claimed that the increased trend in sensitivity of drylands is not due to increased variability. This analysis, while interesting, skipped over the more basic potential cause, that a decrease in precipitation amounts is driving these systems to be more sensitivity. How do we know that points aren't just shifting left (in their figure F1c) due to drier climates and the shift of drylands to become drier (and similarly the wet locations to get wetter) means they will be more sensitive. The authors address this in part later with the multi-model study (but these results are confused by comparing different model configuration/tests) and no justification is given that adding time-varying CO₂ actually improved the models ability to predict LAI. The same applies to the minimalistic model, does the addition of the stomatal closure and LAI increase scaling factors lead to any improvement in how this model represent reality? These additions are really quite interesting, but completely unvalidated (e.g. why is 'a' set to 4 L556, or why is the value for 'b' taken from Australia applicable worldwide L553?)

Overall, the noteworthy result of this paper is that increasing trends in vegetation sensitivities to precipitation are due to CO₂ effects and not climate, land use, or nitrogen deposition and is significant to the field. This conclusion is based on the multi-model intercomparisons, though no justification that varying CO₂ and the other forcing factors actually improves their ability to recreate the observed LAI dataset is given. Again a similar point could be made about the modifications to the minimalistic model. Maybe this has been done by the creators of these models, if not the authors have the LAI datasets on hand to validate this (at least for the 3 models that have all four scenarios in S13). Simplifying the presentation and associated presented mythology for the first part of the manuscript should open space later to better describe and discuss the result that CO₂ concentration variations drive the increasing trend in sensitivity.

L86 You use Aridity as E_p/P in the methods but switch here why. P/E_p is a measure of wetness not aridity, it becomes larger under more moist conditions.

L94 Shouldn't this be Aridity Index >0.65 since aridity is defined as E_p/P later?

L101 So water limited regions are more sensitive to water. This is broadly consistent with many earlier finding dating back to Budyko and the early concepts of light and water limited ecosystems.

L114 The units on this figure are very small and thus difficult to tell how meaningful they are. Can these be recast into something more digestible like % change or something similar.

L122 Feel like you're missing a ref to Fig 2a somewhere here.

L125-128 Consider moving this to the methods

L159 Again this matches with a Wet-Gets-Wetter, Dry-Gets-Drier shift (which is much studied) and as systems get drier they become more sensitive (as in your F1c).

L165 I would think the timescale of response of GRACE is too slow to be meaningful here.

L167 I'm not sure I'm convinced of that this much aggregation is useful. At continental scales there are too many different regions to expect any significant trends. Similarly lumping the world into dryland and not likely smooths out many trends.

L177 Provide justification that the addition of these physical processes to these models improves the representation of the processes that are examined in this study. Does the addition of CO2 improve the models ability to estimate LAI. As shown in F3 (and better in S13) adding time varying CO2 influences the models but how do we know this is better?

L194 Great figure, nice and clean. However, shouldn't the four bars in the 'non-dryland' section total to the blue bar in the 'MEM; section. A quick set of boxes drawn on my PDF tells me these are not consistent. The comparison between these is really not consistent and S13 is a much more clear presentation of this. Note this is a better unit for F1.

L213-216 I suggest rewriting this section to remove the fractions with partials and instead describe things more descriptively.

L215 Again, this shows that a shift in P is a main driver of what's happening.

L235 As you set things up in the equation above (no number, but I think its 1), I was expecting three effects to be: $\partial LAI/\partial T$, $\partial T/\partial ET$, and $\partial ET/\partial P$

L265 What about CO2s effect on climate change driving shifts in P (as shown in S14)

L362 Are you missing the coefs in the autocorrelation model?

L503 Aridity is defined as P/E_p in the results (L86)

L528 There are many other indirect effects (e.g. shifts in soil moisture due to stomatal open less) why focus on this particular direct effect.

L550 Is this eq. missing a negative sign?

L553. I don't follow the jump from 19% to 0.38

Reviewer #2 (Remarks to the Author):

Dear Authors,

You address how vegetation greenness sensitivity to precipitation changes during the last decades, what might be driving those changes and what may be the potential physiological mechanisms. I find the overall ideas interesting, but I think many areas of the manuscript need improvement.

1) It does not feel logical to me to evaluate the drivers of precipitation sensibility directly through outputs of land surface models. I see the value in this, but I would rather see, before this analysis, a simpler one trying to partition the trends to real climate data.

2) The overreliance on modelled data to explain drivers and mechanisms are troubling. How do you separate whether the trends found are true mechanisms/drivers or reflect model assumptions? I do find the analysis valuable, particularly the one for drivers (if paired with a simpler climate data exploration), but I would rather see a better discussion of what the literature has on mechanisms than the ecohydrological model.

3) The aridity index is central to your work, but this index is not discussed in a more clear manner. I find to simplistic putting together all the world's vegetation in a single relative axis. If the relative aridity index is calculated as E_p/P (#503), then that means the same aridity index can be obtained for an area where the actual, non-relative, water deficit is huge as well as for another one where both the water deficit and precipitation are not. I would rather see a non-relative aridity index or, if a relative aridity index is used, that the analysis has precipitation or biomes as a

cofactor. I also miss a map showing which areas have which aridity index.

4) I find the discussion on precipitation sensitivity simplistic. For plants, what matters most is not precipitation, is water availability. Water availability is of course hard to measure, and a non-linear function of precipitation, which is why using sensitivity to precipitation, as a proxy for water availability, has value. But the introduction and discussion have to come back to the actual water availability. Also, please note the trends tested situate into a larger theory which proposes a climatic safety margin to climate change, where wetter places have excess water margin to not be so much affected by precipitation anomaly.

Plant water availability is strongly influenced by soil traits. Using precipitation as a proxy for plant water availability, at global scale, should be ok as long as soil traits are randomly distributed across vegetation types and climate types – thus any local bias becomes random error. I suppose this is acceptable (although a good portion of the Amazon occurs near the water table, thus decoupled from precipitation), but some discussion on this might be interesting.

5) I am not sure how much I trust cloud cover as a proxy for radiation. Also, I would expect cloud cover and precipitation to not be independent. I am not sure how variable dependence is treated in your analysis.

6) Throughout the work, there is an implicit assumption that plants operate by maximizing water use efficiency. This assumption is also behind the models used. This is not true for all environments, particularly for environments where competition for water might be important- in those environments if plants save water for later use, another plant might simply use this water before. The works in the long-term rainfall exclusion experiments have been insightful on this (see Caxiua's experiment in the Amazon). The results of those experiments should be better explored in the introduction and discussion.

7) The results are hard to follow. I needed to go through the methods 2 times slowly to then be able to understand what you did.

Best regards,

Specific comments

#48-49: why even? Do you mean "even within", as for diversity and function turnover within the same forest?

#54-58: and soil traits; the most important precipitation effect is through soil water availability and not precipitation per se. I think it is better if you explicitly construct the theory linking to vegetation responses coupling to water availability, which is mostly driven, in a non-linear fashion, by precipitation.

#58-59 – and acclimation

#61: temporally as in the time domain or momentarily?

#68: of uncertainty for...

#70-80: too much focus on DLM, I could not understand what you did before reading the methods and results once. The DLM is just for getting your main index, which, as far as I understood, you 1) evaluate against an aridity index (no introduction to this anywhere); 2) try to explain drivers using LSMs and 3) try to explain mechanisms using ecohydrological model.

#72-75: the importance here is not the prediction itself, but the model's capability to separate the slope of NDVI ~ precipitation from autocorrelation and other drivers and to estimate this slope over time

#84: "vegetation sensitivity", here and throughout the text – too generic, specify which aspect of vegetation function you are capturing with your data

As your results focus on dry vs wet, I miss a better introduction of the aridity index used. Is the aridity index calculated as E_p/P (line 502?). I would expect an aridity index in the form of actual evapotranspiration divided by potential evapotranspiration.

#89: Fig. S1: what are the lines in e and f?

#129: with a rho of 0.49 only I would hardly the trends similar

#262: plant and vegetation – physiological mechanisms are likely to be different from community level mechanisms and ecosystem level mechanisms. The way the effects are discussed in the following lines are too simplistic.

#281: increased assimilation may also i) be routed to reproductive organs; ii) be stored; iii) be used to change tissue allocation.

#282: eCO₂ only leads to water saving if plants optimize for water use efficiency. This is not true in many ecosystems in which competition for water (and nutrients in the water) are important – in

those systems if one plant does not transpire this water is consumed by other plant.

#332-334: how does the model behave after removing datapoints if it is temporally dependent (t and t-1 in equation 2)?

#362: as the coefficients are not in the equation, maybe mark it as "function of" (f(dNDVI) + f(dPREC)) to be correct. I miss a note on the format of the error term.

#361: what does the delta stands for? Is it just to represent the deseasonalized+detrended? If so, note it in the text.

#364: anomaly is for cloud fraction only, not for precipitation (anomaly)? Anomaly is the difference from the whole time series mean?

#384: was the pixel (grid cell) size noted somewhere?

#384: should you not use the cloud fraction of the preceding month? For sure the NDVI state in any given time is a function of the previous, and not immediate, light availability.

#384: the DLM model is rather complex. I wonder how predictor independency may affect the results. Do precipitation anomaly and cloud fraction anomaly (I suppose they should)? If yes, how do you separate the effects.

#384-387: so you fitted equation 2 using a DLM?

#417-419: it is not immediate clear why you divide in two. You can be more clear you are comparing the sensitivity from period 1 with sensitivity from period 2. I like this analysis, it is simple and conservative, thus robust; the average reader will understand what was done without problem, while the DLM is not trivial but probably much more sensible.

#443 MLR – this only appear two time in the text, no need for acronyms.

#443 – why here you calculate by decade and above you divide the period in two? Maybe keep the same approach in both of them?

#445 – what do you mean by normalized? I imagine it was a scaling or centralization and not a normalization process. Did you used a z-score approach? Would it not be better to say you compared the variances after scaling centralizing?

Reviewer #3 (Remarks to the Author):

Comments to NCOMMS-21-49872

Zhang et al 's work tried to present changes in dryland ecosystems sensitivity to precipitation and explore the potential mechanism behind. While many similar studies have been done in this field, Zhang's work further advanced our understanding of the underlying mechanism of the rising differences of ecosystem response to climate change. The work was done nicely, insightfully and written quite well. However, this manuscript is very long and is easy to get me lost, I would suggest to shorten or discard some general descriptions, such as L206-216 and L416-429. Below is a subset of my concerns and note that the authors may flexibly modify some comments just in case I'm wrong at some place.

Title

The analysis actually have included both wet regions and drylands, and further, I am not sure if it is so correct to use dryland ecosystem in the title as only vegetation greenness are studied and maybe this could be more specific, like drylands vegetation greenness.

Abstract

L26-28 I think the research gap introduced here is not so correct since many similar studies have been done (e.g., referring to relevant references). My question is that could we say the sensitivity of precipitation determine global /regional vegetation dynamic, as precipitation is not always the necessary driver of vegetation growth, like tropical regions. This also raise another issue, the sensitivity analysis in wet regions, does the change in sensitivity make sense? It could be also related to the equation (2), how much precipitation contribute to vegetation variations when considering NDVI at time of lag-1.

L33 Specify the models used, statistically or physically

L34-38, I like this explanations, they sound very convincing, maybe switch the description of wet and dry regions since the drylands was introduced firstly above.

L58 I think reference 18 doesn't make sense.

L73 Specify the data set and period studied.

74-75 I think you do not remove dry season, actually GIMMS data in dry season have large uncertainty.

L78 I don't think coefficient are calculated at each time step is an appealing strength.

L94 and Fig 1 should be better to add a histogram as inset to indicate how many pixels showing a decrease or increase in sensitivity. It's not so prone to catch some obvious pattern due to scattered pixels.

L118-119 the sensitivity trends were binned by aridity, right? If so, how big is each bin?

L122-123 I think this sentence should be rephrased; increased variability of vegetation greenness is caused by increased precipitation variability if an increase in sensitivity is demonstrated. I actually get lost when reading this part about variability, I just feel it doesn't contribute much to the results.

L206-216 better to move this paragraph to method

L218 A convincing explanation and this analysis is insightful.

L262 relationship?

L317 How do authors think about that the sensitivity is expected to decrease due to more frequent and more server climate extremes, it should be right?

Method

GIMMS data during dry season has poor quality and the temporal coverage should be updated to present.

I just feel that NDVI at previous month could contribute a very large proportion to present NDVI? Do you test it?

Do we need to introduce Section 3-4 in such detailed? I think few readers are interested in reading them.

I like this analysis in L453, but I am not sure what's basis for the separations of LAI/T, T/ET, ET/P I cannot follow from L458 onward, maybe the other reviewers could make suggestions.

**Increasing sensitivity of dryland ecosystems to precipitation due to rising atmospheric
CO₂
NCOMMS-21-49872
Response to Reviewers**

We appreciate the constructive comments from the reviewers and the invitation from the editor to submit a revised version. We carefully followed the reviewers' suggestions to carry out additional analyses and improve our manuscript. Please see below our point-to-point responses in blue text following reviewer comments. All line numbers and figures numbers in this response letter refer to the clean version of the revised manuscript.

Reviewer #1 (Remarks to the Author):

The authors of the submitted study conducted a series of statistical and theory driven analyses of vegetation patterns (1981-2015) and their response to vegetation. Dynamic and static, linear and non-linear empirical models are used to show that over the study period vegetation (as measured by satellite NDVI) is more sensitive to precipitation in drylands, with this sensitivity increasing more in drylands. These are complemented with analyses of a suite of terrestrial biosphere models as well as an update to a minimalistic model. These three approaches are all fairly consistent with each other. The authors then evaluated the drivers to the increased trend in sensitivity in drylands in the biosphere and minimalistic models and concluded that these changes are due to the role of CO₂ on vegetation.

Response: We appreciate the reviewer's positive comments.

It is very clear the authors have done a huge amount of work investigating the stated problem. The paper is fairly well written, however the overabundance of methods and approaches employed somewhat obscures the findings. For instance, the univariate and multivariate methods produce similar results (L102) as do the MLR and DLR produce similar results (L88). This suggests that similar conclusions can be drawn with simple methods. Additionally, the comparison between the different models in F3 mixes and matches and their presentation and isn't consistent (F13 is much more-clear in this regard). The section on the minimalistic modeling is interesting (and could be its own theoretical paper for instance) and should be made more descriptive by removing the partial derivative terms from the text where possible and instead reframing these processes by descriptive names. I suggest the authors clean up and simplify the methods presented (perhaps moving more to the supplementary) in order to create more space for improve descriptions and discussions. In fact much of the first part of the results (the finding that water limited ecosystems are more sensitive to water limitations) could be compressed to focus on the drivers of this.

Response: We agree with the reviewer that a lot of methods are involved in our study and we have made a lot of improvements to the structure of the presentation in this revision. First, we curtailed the first section of the results, i.e., the finding of the precipitation sensitivity trend. We decided to keep the DLM results only and moved the robustness tests and corresponding discussion to the supplementary discussion section. The reason we did not use the MLR is that DLM can provide a time-varying precipitation sensitivity, which is important for characterizing the sensitivity changes over time for different regions (Fig. S10) and estimating the trend of NDVI variations due to precipitation (Fig. 2). Second, we appreciate the reviewer's suggestion on using Figure S13 (now Figure S15 in the revised manuscript) as the figure in the main text, but considering Figure S13 only uses three models and the results are qualitatively similar for both

figures, we feel it might be better to keep Figure 3 as is. Nevertheless, we agree this mismatch may be confusing to readers and we added explanations in the figure caption (L185). Third, we revised the text and avoided the partial derivative terms in the results section. Fourth, we moved detailed description of the DLM and MLR method into the supplementary information. These changes have allowed us to focus more on the drivers of the precipitation sensitivity trend. Please also refer to our responses to your comments below for the detailed description of further changes.

In particular, the finding that when water isn't a limiting factor then ecosystems are not sensitive to water is not particularly new or novel. This is expected based on a simple Budyko analysis from which the authors incorrectly draw their definition of drylands (aridity is PET/P not P/PET). Water limited regions are sensitive to water and light limited regions are sensitive to light, so as they move from light limited to water limited under climate change their sensitivities go up, with stronger increases for more arid drylands (as shown in S12). In the results (and in F2) it is claimed that the increased trend in sensitivity of drylands is not due to increased variability. This analysis, while interesting, skipped over the more basic potential cause, that a decrease in precipitation amounts is driving these systems to be more sensitive. How do we know that points aren't just shifting left (in their figure F1c) due to drier climates and the shift of drylands to become drier (and similarly the wet locations to get wetter) means they will be more sensitive. The authors address this in part later with the multi-model study (but these results are confused by comparing different model configuration/tests) and no justification is given that adding time-varying CO_2 actually improved the models ability to predict LAI. The same applies to the minimalistic model, does the addition of the stomatal closure and LAI increase scaling factors lead to any improvement in how this model represent reality? These additions are really quite interesting, but completely unvalidated (e.g. why is 'a' set to 4 L556, or why is the value for 'b' taken from Australia applicable worldwide L553?)

Response: We appreciate the reviewer's summary for the results and would like to clarify several key issues.

1) The definition of the aridity index is corrected based on conventional use. In the literature both of those definitions have been used but for readability we use P/PET as aridity index throughout (dryland is also based on this definition). Please refer to our response to your comment to L86 below.

2) In Figure S13 (Figure S15 in the revised SI) and other parts in the results (e.g., Fig. S2), we postulate that the contrasting trend in sensitivity is not primarily caused by precipitation changes. This is because, a) there is no significant trend in precipitation for most continents except for Africa. b) although as the reviewer suggests, the sensitivity is expected to change when an ecosystem is getting drier or wetter, we find that in Figure S13 when there is no precipitation trend ($y\text{-axis}=0$), the contrasting trend in precipitation sensitivity between dry and wet still exist. This implies that changes in precipitation may not explain the contrasting trend in precipitation sensitivity between dry and wet. This is further validated by our factorial modeling analysis (Fig. 3). We clarified these issues in the revised manuscript (L140-147)

3) We used the Terrestrial Biosphere Models from MsTMIP to explore the cause for the contrasting trends of precipitation sensitivity between dry and wet. We did not validate the models' capability in representing LAI against observations, nor whether or not the model prediction of LAI is improved when CO_2 is included. This is because the LAI variations are mostly contributed by the variations in the spatial and seasonal domain. The CO_2 is known to have limited influence at these domains. Adding CO_2 effect into the model is not likely to significantly improve the model

performance. However, we did compare the LAI sensitivity to precipitation, which is the focus of our study. The results show that when CO₂ is included, models predict a contrasting trend of the sensitivity, while with other factors (varying climate, land use change, nitrogen deposition), models cannot reproduce such trends or the magnitudes do not match (Fig. 3). Admittedly, there could still be some discrepancies between model simulations and observations. The argument is that if the major processes are included in the model (for example, CO₂ fertilization effect on photosynthesis, water limitation on stomatal conductance and carboxylation, carbon allocation, ecosystem water balance, etc.), the models can predict the correct response to different environmental driving factors, allowing us to separate the respective contribution of each factor. This is a commonly used approach in the modeling community (for example, Zhu et al., 2016), and while subject to uncertainty it has proven powerful for inference.

4) We agree that we did not validate the minimalistic model against LAI observations, but in the paper when the model is first proposed, Good et al. validated the models' capability in representing the spatial patterns of T/P against multiple observation & modeling datasets (Good et al., 2017). In our study, it can be seen that the general pattern of precipitation sensitivity matches with what we get from the satellite observations (Figure S6). In fact, we do not expect this simple model to fully reproduce the LAI trend or variability, rather, it is a simple way to validate and explore the mechanisms based on our current knowledge. We agree that the model parameterization may not perfectly represent the real world because of the embedded uncertainties. In this revision, we also tested several different parameter combinations, and the results show that the using different *a* values (3,4,5) and *b* values (0.38, 0.2, 0.5) do not change our conclusion (Fig. R1). We have added this figure in the supplementary information and discussed the effect of using other parameters.

Fig. R1. Predicted responses of the changes in LAI to precipitation sensitivity along the aridity index. Different line types and colors correspond to different *a* and *b* combinations.

Zhu, Z., Piao, S., Myneni, R.B., Huang, M., Zeng, Z., Canadell, J.G., Ciais, P., Sitch, S., Friedlingstein, P., Arneeth, A., Cao, C., Cheng, L., Kato, E., Koven, C., Li, Y., Lian, X., Liu, Y., Liu, R., Mao, J., Pan, Y., Peng, S., Peñuelas, J., Poulter, B., Pugh, T.A.M., Stocker, B.D., Viovy, N., Wang, X., Wang, Y., Xiao, Z., Yang, H., Zaehle, S., Zeng, N., 2016. Greening of the Earth and its drivers. *Nature Climate Change* 6, 791–795. <https://doi.org/10.1038/nclimate3004>

Good, S.P., Moore, G.W., Miralles, D.G., 2017. A mesic maximum in biological water use demarcates biome sensitivity to aridity shifts. *Nature Ecology & Evolution* 1, 1883. <https://doi.org/10.1038/s41559-017-0371-8>

Overall, the noteworthy result of this paper is that increasing trends in vegetation sensitivities to precipitation are due to CO₂ effects and not climate, land use, or nitrogen deposition and is significant to the field. This conclusion is based on the multi-model intercomparisons, though no justification that varying CO₂ and the other forcing factors actually improves their ability to recreate the observed LAI dataset is given. Again a similar point could be made about the modifications to the minimalistic model. Maybe this has been done by the creators of these models, if not the authors have the LAI datasets on hand to validate this (at least for the 3 models that have all four scenarios in S13). Simplifying the presentation and associated presented mythology for the first part of the manuscript should open space later to better describe and discuss the result that CO₂ concentration variations drive the increasing trend in sensitivity.

Response: We appreciate the reviewer's suggestions. There is a consensus that model predicted LAI still has large uncertainty compared with satellite observed LAI. The effect of CO₂ on LAI is gradual and is more evident at longer timescale. For example, Zhu et al. (2016) show that models can reproduce the satellite observed greening Earth and CO₂ increases contributed largely to this increase. We used 10 state-of-the-art models which differ in terms of model structures, processes included, and parameterization, etc. These models should represent our current knowledge on ecosystem responses to the environment. Again, we would like to highlight that our study does not focus on the LAI spatial or seasonal dynamics, which are the first order variations of LAI; rather, we are interested in the contrasting trend in vegetation sensitivity to precipitation between the dry and wet. This is validated in Fig. 3, where satellite observations and model predictions under that "all-varying" scenario (BG1) show similar trends and magnitudes for the dry and wet regions, respectively. If the CO₂ effect is not considered in the model, the sensitivity trend does not match with the satellite observations. As for the minimalistic model, the model prediction of precipitation sensitivity pattern along aridity index is also similar to the satellite observations (Fig. 4 and Fig. S6).

We have followed the reviewer's suggestions and shortened the first section of the results. We moved most of the robustness tests into the supplementary discussion and associated methods into the supplementary methods. We also added results and discussions related to the modeling analysis (L242-247, L273-275).

L86 You use Aridity as E_p/P in the methods but switch here why. P/E_p is a measure of wetness not aridity, it becomes larger under more moist conditions.

Response: We thank the reviewer for pointing this out. There was a conflict between these two definitions in the previous version of the manuscript. In the revised main text, we followed the conventional definition of aridity index by World Atlas of Desertification and many other studies, and from which dryland is defined by United Nations Environment Programme (UNEP). They define aridity index as precipitation over potential evapotranspiration. In the description of the minimalistic model, we define the inverse of aridity index as the dryness index, which is often used in hydrological studies (e.g., Creed et al. 2014). We did not use an aridity index in the minimalistic model derivation since it would complicate the equations.

Creed, I.F., Spargo, A.T., Jones, J.A., Buttle, J.M., Adams, M.B., Beall, F.D., Booth, E.G., Campbell, J.L., Clow, D., Elder, K., Green, M.B., Grimm, N.B., Miniati, C., Ramlal, P., Saha, A., Sebestyen, S., Spittlehouse, D., Sterling, S., Williams, M.W., Winkler, R., Yao, H., 2014. Changing forest water yields in response to climate warming: results from long-term experimental watershed sites across North America. *Glob Change Biol* 20, 3191–3208. <https://doi.org/10.1111/gcb.12615>

L94 Shouldn't this be Aridity Index >0.65 since aridity is defined as E_p/P later?

Response: We are sorry for this oversight. Please see our response above.

L101 So water limited regions are more sensitive to water. This is broadly consistent with many earlier finding dating back to Budyko and the early concepts of light and water limited ecosystems.

Response: Yes, the spatial pattern of vegetation canopy sensitivity to precipitation is consistent with previous studies. Here we highlight the trend is increasing for the drylands and decreasing for the non-drylands.

L114 The units on this figure are very small and thus difficult to tell how meaningful they are. Can these be recast into something more digestible like % change or something similar.

Response: The unit is small since NDVI is unitless and is between 0-1, the de-seasonalized detrended NDVI anomalies which is used for the sensitivity calculation is often 1-2 orders of magnitude smaller (0.01). The range of precipitation anomaly is mostly between 10-200 mm, these large differences lead to this small number in absolute value. We agree that the extreme small numbers are difficult to interpret its physical meaning, we therefore changed the unit (NDVI $m^{-1} H_2O$) and they are now in a more interpretable range. We also changed other figures throughout the manuscript.

L122 Feel like you're missing a ref to Fig 2a somewhere here.

Response: Thank you for pointing this out, we added the citation for Fig. 2a in Line 116.

L125-128 Consider moving this to the methods

Response: We followed the reviewer's suggestion and moved this part to the method section, considering the importance of $\sigma NDVI_{prec}$ in understanding Fig. 2, we also briefly explained $\sigma NDVI_{prec}$ in the figure legends.

L159 Again this matches with a Wet-Gets-Wetter, Dry-Gets-Drier shift (which is much studied) and as systems get drier they become more sensitive (as in your F1c).

Response: We agree, this is another possible explanation why we would see a contrasting trend in precipitation sensitivity between dry and wet. However, the WWDD paradigm is originally proposed for oceans and previous studies do not support this hypothesis over land (Held and Soden 2006; Greve et al., 2014; Xiong et al., 2022). Considering we did not find obvious precipitation trend for the dry and wet (Figure S2), this hypothesis is not supported in terms of precipitation. We did another independent test in the Figure S13. Along precipitation trend (y-axis) and aridity index (x-axis), when there is no obvious trend for precipitation ($y=0$), precipitation

sensitivity increases for the dry and decreases for the wet. This suggests water availability may not be driven by precipitation change, but other factors.

Greve, P., Orlowsky, B., Mueller, B., Sheffield, J., Reichstein, M., Seneviratne, S.I., 2014. Global assessment of trends in wetting and drying over land. *Nature Geoscience* 7, 716–721. <https://doi.org/10.1038/ngeo2247>

Held, I.M., Soden, B.J., 2006. Robust Responses of the Hydrological Cycle to Global Warming. *Journal of Climate* 19, 5686–5699. <https://doi.org/10.1175/JCLI3990.1>

Xiong, J., Guo, S., Chen, J., Yin, J., 2022. A reexamination of the dry gets drier and wet gets wetter paradigm over global land: insight from terrestrial water storage changes. *Hydrol. Earth Syst. Sci. Discuss.* <https://doi.org/10.5194/hess-2021-645>

L165 I would think the timescale of response of GRACE is too slow to be meaningful here.

Response: GRACE TWS has shown to be strongly coupled to land carbon sink at monthly time scale (Humphrey et al., 2018), there could be a potential linkage between the trend in TWS and the trend in precipitation sensitivity. We agree with the reviewer that the signal from GRACE has some limitations, we have removed this analysis in the revised manuscript. The new subplot shows θ_{prec} in the AI-PET trend 2D space (Fig. S13), but with no obvious pattern along the PET trend axis.

Humphrey, V., Zscheischler, J., Ciais, P., Gudmundsson, L., Sitch, S., Seneviratne, S.I., 2018. Sensitivity of atmospheric CO₂ growth rate to observed changes in terrestrial water storage. *Nature* 560, 628–631. <https://doi.org/10.1038/s41586-018-0424-4>

L167 I'm not sure I'm convinced of that this much aggregation is useful. At continental scales there are too many different regions to expect any significant trends. Similarly lumping the world into dryland and not likely smooths out many trends.

Response: We agree this aggregated trend cannot be treated as direct evidence. We therefore revised the sentence and it now reads:

“However, because precipitation trends are not significant in both drylands and non-drylands, precipitation change alone cannot explain the contrasting trends in θ_{prec} (Supplementary Fig. S2c).”

L177 Provide justification that the addition of these physical processes to these models improves the representation of the processes that are examined in this study. Does the addition of CO₂ improve the models ability to estimate LAI. As shown in F3 (and better in S13) adding time varying CO₂ influences the models but how do we know this is better?

Response: Please refer to our response to your major comments #2 and #3.

L194 Great figure, nice and clean. However, shouldn't the four bars in the 'non-dryland' section total to the blue bar in the 'MMEM; section. A quick set of boxes drawn on my PDF tells me these are not consistent. The comparison between these is really not consistent and S13 is a much more clear presentation of this. Note this is a better unit for F1.

Response: The reviewer is correct, in Figure 3, the total effects for each factor (CO₂, climate, etc.) do not add up to the MMEM. This is due to the fact that in Figure 3, the effect for each factor is calculated from the average of all available pairs of model simulations. For example, if model A,B,C provide simulations for SG1,SG2 scenarios (from which the CO₂ effect can be calculated), but only model A,C provide simulations for SG1 and BG1 (from which the MMEM can be calculated), then model B only contributes to the calculation of the CO₂ not the MMEM. For Figure S13, the MMEM and contributions from all factors are calculated from the same set of models. As expected, limited (only three) models are used for Fig. S13. This may lead to relatively larger uncertainties. We therefore decide to keep Fig. 3 as is. But we also added explanations of the mismatch in the figure caption.

L213-216 I suggest rewriting this section to remove the fractions with partials and instead describe things more descriptively.

Response: Thank you for the suggestions, we rewrite the section with more descriptive words and minimize the partial derivative notions being used. (L192-196)

L215 Again, this shows that a shift in P is a main driver of what's happening.

Response: We cannot get solid evidence from this equation, since we are focusing on the partial derivative, not the ratio between ET and P. The shift in P can drive the change in ET/P, but it is not very clear how that would change the partial derivative. We agree that precipitation plays an important role and directly affect the sensitivity changes, but here we are focusing on the contrasting trend between dry and wet, which cannot be explained by the precipitation changes.

L235 As you set things up in the equation above (no number, but I think its 1), I was expecting three effects to be: $\partial_{LAI}/\partial_{T}$, $\partial_{T}/\partial_{ET}$, and $\partial_{ET}/\partial_{P}$

Response: We have examined the CO₂ effect on these three components in the factorial model simulations (Fig. S16). However, through those analyses, it is still unknown what mechanism drives the changes of these three components, and why do they respond differently to the dry and wet. By using this minimalistic model, we can identify two major CO₂ effects, i.e., the stomatal closure, and enhanced LAI in drylands, and how they affect the components above. This is a further step after the factorial analysis. We explained this in the revised manuscript.

“Using this simple model, we can decompose the sensitivity changes into three different CO₂ effects, and understand what mechanism drives the contrasting trends between dryland and non-dryland.”

L265 What about CO₂'s effect on climate change driving shifts in P (as shown in S14)

Response: It is not very clear that how CO₂ can change the precipitation, especially over land. In our study, since observed precipitation is directly used in the DLM, we do not consider the CO₂ effect on precipitation to be an indirect effect.

L362 Are you missing the coeffs in the autocorrelation model?

Response: We have rewritten the Eq (2) and (3) as an equation form with coefficients.

L503 Aridity is defined as P/E_p in the results (L86)

Response: Thank you for pointing this out. Here we define E_p/P as a “dryness index”, which is often used by hydrological studies. The aridity index is still defined as P/E_p in the main text.

L528 There are many other indirect effects (e.g. shifts in soil moisture due to stomatal open less) why focus on this particular direct effect.

Response: Since the original minimalistic model is capable to reproduce the multiple key hydrological process (e.g., interception, soil moisture dynamics, evaporation and transpiration), here we only focus on the CO_2 effect on plants. The change in stomatal conductance and change in LAI is the most important CO_2 effect on plants. Other effects, like the reviewer suggest, can be facilitated through the combination of the original model and the modified effects on plants.

L550 Is this eq. missing a negative sign?

Response: Here we use the positive side of the monotonical decreasing sigmoid function (from dry to wet) (Figure Rxa).

Fig R2. (a) Two types of sigmoid function, (b) the function used in our study.

L553. I don't follow the jump from 19% to 0.38

Response: Since we only use half of the sigmoid function, the range is between 0 and 0.5. We therefore need to double b so that ζ equals 19% at the dry end. See Figure R2b. We added explanations about this in the revised manuscript:

“We therefore set b as 0.38 so that ζ is in the range of (1, 1.19].”

Reviewer #2 (Remarks to the Author):

Dear Authors,

You address how vegetation greenness sensitivity to precipitation changes during the last decades, what might be driving those changes and what may be the potential physiological mechanisms. I find the overall ideas interesting, but I think many areas of the manuscript need improvement.

Response: Thank you for the positive comments, we have carefully revised the manuscript based on your suggestions.

1) It does not feel logical to me to evaluate the drivers of precipitation sensibility directly through outputs of land surface models. I see the value in this, but I would rather see, before this analysis, a simpler one trying to partition the trends to real climate data.

Response: We fully agree with the reviewer—we did this analysis but perhaps we were not clear in the presentation of our results. Considering that there is a strong change along the aridity index gradient (Fig. 1C), one possible assumption may be due to the climate induced change of aridity. For example, as the sensitivity almost monotonically decrease with the aridity index, a drying trend in the dryland together with a wetting trend in the non-dryland would also lead to such contrasting trends. We tested such hypothesis by analyzing the precipitation trend for each continent (Fig. S2) and analyzed the θ_{prec} trend for those pixels with no obvious changes of precipitation (Figure S13). Both analyses do not support this hypothesis. Here we followed the reviewer's suggestion and tested the relationship between θ_{prec} trend and the trends of precipitation, temperature and cloud cover. We found very weak relationship between the sensitivity trend and trend of all climate variables (Figure R3). And there is no significant difference between dryland and non-dryland. This suggests that the contrasting trend between dry and wet is not likely to be attributed to the changes in climate variables. We added this analysis into the supporting information (Figure S14) and discussed this in the main text (L146-147).

Fig. R3. Relationship between trend of precipitation sensitivity and trend in climate variables (precipitation, temperature, cloud cover). The first row shows the correlation for all pixels (a-c), second row for the dryland pixels (d-f), third row for the non-dryland pixels (g-i).

2) The overreliance on modelled data to explain drivers and mechanisms are troubling. How do you separate whether the trends found are true mechanisms/drivers or reflect model assumptions? I do find the analysis valuable, particularly the one for drivers (if paired with a simpler climate data exploration), but I would rather see a better discussion of what the literature has on mechanisms than the ecohydrological model.

Response: We did several tests in the manuscript to explore the possible mechanism and disentangle model assumptions from underlying mechanisms. Based on figure 1c, it is expected that a drying trend or wetting trend (change in precipitation) can induce changes of precipitation

sensitivity. However, our analysis shows no obvious precipitation trend for dryland and non-dryland (Figure S2). Further analysis shows that for regions with no trend in precipitation, the contrasting trends in precipitation sensitivity still exist (Figure S13). These suggest that precipitation or climate changes is not likely to be the cause. This is also confirmed by the MsTMIP models.

In the factorial modeling analysis (MsTMIP), we used 10 state-of-the-art models which differ in the model structure, processes included, and parameterization, etc. These models are a representation of a suite of different assumptions and current knowledge on ecosystem responses to the environment. The minimalist model analysis is consistent with the factorial modeling analysis. Both of which can reproduce the contrasting trend we observed from remote sensing dataset. These two lines of evidence suggest that the mechanism we obtained is robust.

As suggested, we also discussed the mechanism based on previous literature in the first two paragraphs of the discussion section. The direct effect of CO₂ on plants' stomatal conductance and the indirect effect of CO₂ that LAI increase is larger in drylands are both supported by previous literature, either from FACE experiment or through modelling analysis. The combination of these two effects help explain our findings.

3) The aridity index is central to your work, but this index is not discussed in a more clear manner. I find to simplistic putting together all the world's vegetation in a single relative axis. If the relative aridity index is calculated as E_p/P (#503), then that means the same aridity index can be obtained for an area where the actual, non-relative, water deficit is huge as well as for another one where both the water deficit and precipitation are not. I would rather see a non-relative aridity index or, if a relative aridity index is used, that the analysis has precipitation or biomes as a cofactor. I also miss a map showing which areas have which aridity index.

Response: The aridity index is defined as precipitation over potential evapotranspiration, it is an important metric in the ecohydrology studies which directly related to the vegetation functioning (Lian et al., 2021; Berdugo et al., 2020; Fatichi et al., 2016), and ecosystem water partitioning into evaporation, transpiration, and runoff through the Budyko framework (Budyko, 1974; Good et al., 2017). The reviewer is correct that a same aridity index can be achieved with different combinations of PET and P values. However, as shown in Figure R4, the spatial variation of PET (coefficient of variation, $cv=0.29$) in our study area is much smaller than that of P ($cv=0.67$), suggesting that much of the spatial variation of aridity index is attributed to P.

One unique advantage of this metric is that it describes the relative relationship between the water supply and demand, and is a good indicator to show how strong water is limiting for the ecosystem. Considering that our study focuses on the precipitation sensitivity, it is straightforward to link the sensitivity to how strong the ecosystem is limited by water (i.e., the aridity index). Since precipitation is directly used for the aridity index calculation, and there is a strong relationship between the aridity index and ecosystem canopy coverage, it is a more appropriate index for our study on dryland and non-dryland.

We added a map of aridity index in the supplementary information (Figure R5 or Figure S1), and add additional description of aridity index in the methods (L339-342).

Fig. R4. Spatial patterns of the mean annual potential evapotranspiration (a) and mean annual precipitation (b). Insets show the histogram of the PET and P. The units are mm/year.

Fig. R5. The map of the aridity index (P/PET) for the study region. Colors from red to green correspond to hyper-arid, arid, semi-arid, dry sub-humid, and humid, respectively. Dryland corresponds to areas with an aridity index smaller than 0.65 (i.e., red and yellow color). White area represents barren land with no vegetation.

Berdugo, M., Delgado-Baquerizo, M., Soliveres, S., Hernández-Clemente, R., Zhao, Y., Gaitán, J.J., Gross, N., Saiz, H., Maire, V., Lehman, A., Rillig, M.C., Solé, R.V., Maestre, F.T., 2020. Global ecosystem thresholds driven by aridity. *Science* 367, 787–790. <https://doi.org/10.1126/science.aay5958>

Budyko, M.I., 1974. *Climate and life*. Academic press.

Good, S.P., Moore, G.W., Miralles, D.G., 2017. A mesic maximum in biological water use demarcates biome sensitivity to aridity shifts. *Nature Ecology & Evolution* 1, 1883. <https://doi.org/10.1038/s41559-017-0371-8>

Fatichi, S., Leuzinger, S., Paschalis, A., Langley, J.A., Donnellan Barraclough, A., Hovenden, M.J., 2016. Partitioning direct and indirect effects reveals the response of water-limited ecosystems to elevated CO₂. *Proceedings of the National Academy of Sciences* 113, 12757–12762. <https://doi.org/10.1073/pnas.1605036113>

Lian, X., Piao, S., Chen, A., Huntingford, C., Fu, B., Li, L.Z.X., Huang, J., Sheffield, J., Berg, A.M., Keenan, T.F., McVicar, T.R., Wada, Y., Wang, X., Wang, T., Yang, Y., Roderick, M.L., 2021. Multifaceted characteristics of dryland aridity changes in a warming world. *Nat Rev Earth Environ* 2, 232–250. <https://doi.org/10.1038/s43017-021-00144-0>

4) I find the discussion on precipitation sensitivity simplistic. For plants, what matters most is not precipitation, is water availability. Water availability is of course hard to measure, and a non-linear

function of precipitation, which is why using sensitivity to precipitation, as a proxy for water availability, has value. But the introduction and discussion have to come back to the actual water availability. Also, please note the trends tested situate into a larger theory which proposes a climatic safety margin to climate change, where wetter places have excess water margin to not be so much affected by precipitation anomaly.

Plant water availability is strongly influenced by soil traits. Using precipitation as a proxy for plant water availability, at global scale, should be ok as long as soil traits are randomly distributed across vegetation types and climate types – thus any local bias becomes random error. I suppose this is acceptable (although a good portion of the Amazon occurs near the water table, thus decoupled from precipitation), but some discussion on this might be interesting.

Response: We appreciate the reviewer's insights. We agree that plants are directly affected by the soil water availability, and the soil water is not only affected by the precipitation water supply and runoff, but also affected by the dynamic water consumption by plants. The water saving effect in the wet regions also indicates a greater excess water margin, this is consistent with the decreasing precipitation sensitivity in the non-dryland area. Actually, the minimalistic model is capable of predicting such decoupling relationship (Fig. 4a), since soil water-precipitation relationship is well represented in the model. We followed the reviewer's suggestions and discussed the importance of soil water availability in regulating the vegetation in both the introduction and discussion (L58-61, L244-249).

The spatial heterogeneity is an important issue for our study, especially when considering the soil traits that directly affect the soil water dynamics and plant traits that regulates the plant water use. These variations are not considered in our study because 1) as the reviewer suggested, their distribution may be random, 2) we assume these traits to be static and cannot explain the temporal changes of precipitation sensitivity. We also discussed these in the revised manuscript (L275-277).

5) I am not sure how much I trust cloud cover as a proxy for radiation. Also, I would expect cloud cover and precipitation to not be independent. I am not sure how variable dependence is treated in your analysis.

Response: We tested the correlation between cloud cover and radiation, cloud cover and precipitation (Figure R6). There is a strong negative correlation (-0.81 ± 0.13) between cloud and radiation, and a weak positive correlation between cloud cover and precipitation (0.28 ± 0.17). We removed the mean seasonal cycle for all three variables before we calculate the correlation.

Input variables should be independent from each other for the multivariate regression and the dynamic linear models. The collinearity between the input variables may induce changes to the sensitivity factor. We therefore tested both the univariate model which only include the lag-1 autocorrelation term and precipitation, and the multivariate model which consider other climate variables (i.e., cloud cover and temperature). Both models show very similar results. We are therefore confident that our finding has statistical foundation.

Figure. R6. (a) the correlation between the de-seasonalized anomalies of cloud cover and shortwave radiation. (b) the correlation between the de-seasonalized anomalies of cloud cover and concurrent precipitation. The radiation is from CRU-NCEP.

6) Throughout the work, there is an implicit assumption that plants operate by maximizing water use efficiency. This assumption is also behind the models used. This is not true for all environments, particularly for environments where competition for water might be important- in those environments if plants save water for later use, another plant might simply use this water before. The works in the long-term rainfall exclusion experiments have been insightful on this (see Caxiuana's experiment in the Amazon). The results of those experiments should be better explored in the introduction and discussion.

Response: We appreciate the reviewer's insights. The optimization for plant water use have been long studied and multiple stomatal conductance models have been proposed. However, as the reviewer suggests, these models do not consider the competition between individuals, which is very important for most dryland ecosystems. Recent advances try to incorporate plant hydraulic theory into these stomatal conductance models so that the effect of competition can be better represented. Wolf et al. (2016) proposed a new optimization target considering the competition of water resource between individuals. The effect of competition on individual plant growth also depends on the types of competition (intraspecific or interspecific), species characteristics and local environment. For example, interspecific competition may increase growth for certain species at non-drought-prone environment, while intraspecific competition decrease growth for most cases (González de Andrés et al., 2018). Hydraulic diversity also influences the ecosystem responses to precipitation anomalies (Anderegg et al., 2018), with higher diversity showing stronger resistance to environmental anomalies. We added these to the discussion (L242-249).

Wolf, A., Anderegg, W.R.L., Pacala, S.W., 2016. Optimal stomatal behavior with competition for water and risk of hydraulic impairment. *Proceedings of the National Academy of Sciences* 113, E7222–E7230. <https://doi.org/10.1073/pnas.1615144113>

Anderegg, W.R.L., Konings, A.G., Trugman, A.T., Yu, K., Bowling, D.R., Gabbitas, R., Karp, D.S., Pacala, S., Sperry, J.S., Sulman, B.N., Zenes, N., 2018. Hydraulic diversity of forests regulates ecosystem resilience during drought. *Nature* 561, 538–541. <https://doi.org/10.1038/s41586-018-0539-7>

González de Andrés, E., Camarero, J.J., Blanco, J.A., Imbert, J.B., Lo, Y.-H., Sangüesa-Barreda, G., Castillo, F.J., 2018. Tree-to-tree competition in mixed European beech-Scots pine forests has different impacts on growth and water-use efficiency depending on site conditions. *J Ecol* 106, 59–75. <https://doi.org/10.1111/1365-2745.12813>

7) The results are hard to follow. I needed to go through the methods 2 times slowly to then be able to understand what you did.

Response: We have revised and simplified the manuscript and reorganized the results section. For example, most of the description regarding to the robustness of the contrasting trend between dry and wet is moved to the supplementary discussion. The description of Figure 2 is also simplified, it now only provides the most important information on NDVI variability induced by precipitation. We also modified section where the factorial modeling analysis is used to understand the CO₂ effect on precipitation sensitivity.

We also realized that the methods may provide too many details and dilutes the key ideas behind the methods. We now briefly introduced the concept of the DLM and multi-variate regression method. The detailed information is moved to the supplementary method section.

With these changes made, we hope the manuscript is now easier for the readers to follow.

Best regards,

Specific comments

#48-49: why even? Do you mean “even within”, as for diversity and function turnover within the same forest?

Response: This sentence is revised to:

“even in tropical forests which is often not considered water-limited.”

#54-58: and soil traits; the most important precipitation effect is through soil water availability and not precipitation per se. I think it is better if you explicitly construct the theory linking to vegetation responses coupling to water availability, which is mostly driven, in a non-linear fashion, by precipitation.

Response: We appreciate the reviewer’s suggestions. We have added soil texture here. We agree with the reviewer that the soil water availability is the actual driver of the ecosystem functioning. That is also why we incorporate the minimalistic model in the last section of the study. The minimalistic model simulates soil moisture dynamics based on the precipitation frequency, intensity, evaporative demand, and soil characteristics. The soil moisture dynamics further drives the partition of the precipitation to interception, evaporation, transpiration and runoff. This allows us to better understand how CO₂ affect these processes differently.

#58-59 – and acclimation

Response: We agree that plants can acclimate to the environment (e.g., precipitation changes in Schuldt et al., 2011), and this also leads to changes of the plants’ sensitivity to precipitation. We therefore revised the sentence and it now reads:

“Temporally, in addition to the changes in hydroclimate and consequent plants acclimation²⁰, shifts in species composition²¹, factors that lead to changes ...”

Schuldt, B., Leuschner, C., Horna, V., Moser, G., Köhler, M., van Straaten, O., Barus, H., 2011. Change in hydraulic properties and leaf traits in a tall rainforest tree species subjected to long-term throughfall exclusion in the perhumid tropics. *Biogeosciences* 8, 2179–2194. <https://doi.org/10.5194/bg-8-2179-2011>

#61: temporally as in the time domain or momentarily?

Response: Here we mean in the time domain, we realized that we are discussing this under the temporal domain (Line 58), we felt this is not necessary and removed this word.

#68: of uncertainty for...

Response: We have revised this to

“this knowledge gap constitutes a large source of uncertainty in projection of future climate change.”

#70-80: too much focus on DLM, I could not understand what you did before reading the methods and results once. The DLM is just for getting your main index, which, as far as I understood, you 1) evaluate against an aridity index (no introduction to this anywhere); 2) try to explain drivers using LSMs and 3) try to explain mechanisms using ecohydrological model.

Response: The reviewer is correct. We realized that the readers may get confused with such detailed information on the DLM in the introduction. We modified this paragraph by only explaining what can the DLM do and what its strengths are. We also added descriptions on other methods being used in the study (L74-81). This helps the readers to better understand the logic flow of this study.

#72-75: the importance here is not the prediction itself, but the model's capability to separate the slope of NDVI ~ precipitation from autocorrelation and other drivers and to estimate this slope over time

Response: We thank the reviewer for this clarification. The strength of DLM is indeed to get a robust time-varying precipitation sensitivity. We rephrase this sentence into:

“The approach is based on a dynamic linear model (DLM), which can estimate the time-varying relationship between environmental factors and NDVI from GIMMS (Methods), allowing us to derive a robust estimate of precipitation sensitivity with autocorrelation and other climate factors properly considered.”

#84: “vegetation sensitivity”, here and throughout the text – too generic, specify which aspect of vegetation function you are capturing with your data

Response: Thank you for the suggestion, we have specified that we are focusing on the “sensitivity of vegetation canopy greenness to precipitation (θ_{prec})” in the revised manuscript.

As your results focus on dry vs wet, I miss a better introduction of the aridity index used. Is the aridity index calculated as E_p/P (line 502?). I would expect an aridity index in the form of actual evapotranspiration divided by potential evapotranspiration.

Response: We define aridity index as precipitation over potential evapotranspiration in Line 86. Aridity index is a well-established idea which is widely used for hydroclimate studies. Here we follow its conventional definition which is also used by United Nations Environment Programme (UNEP) to define dryland. We have revised the definition of E_p/P as dryness index and added a more detailed description of aridity index in the method section.

#89: Fig. S1: what are the lines in e and f?

Response: We are sorry for this oversight, the lines in e and f indicate the mean and trend of the precipitation sensitivity along the aridity index. We added figure legends for these two subplots.

#129: with a rho of 0.49 only I would hardly the trends similar

Response: We have revised this sentence to:

“The spatial pattern of $\sigma NDVI_{prec}$ trends are more similar to the θ_{prec} trend than the precipitation variability trend (Spearman’s $\rho=0.49$ vs 0.11 , $P<0.001$, Supplementary Fig. S9)”

#262: plant and vegetation – physiological mechanisms are likely to be different from community level mechanisms and ecosystem level mechanisms. The way the effects are discussed in the following lines are too simplistic.

Response: We agree that the plant physiological mechanism may be different at individual level and at ecosystem level. A good example is that, as the reviewer mentioned later, when different individuals compete for water. At individual level, stomatal optimization through a constant marginal water use efficiency may save water for future use. At ecosystem level, the saved water may be used by other more aggressive plants. Two strategies may be adopted, one is that the stomatal use different optimizing strategy, for example, stomata may also respond to the leaf water potential so that a balance can be reached between maximizing water transport and minimizing xylem damage repair (Wolf et al., 2016); the other is that plants may growth may be altered depending on the species characteristics, ecosystem diversity and local environment. We added these aspects in the discussion.

Wolf, A., Anderegg, W.R.L., Pacala, S.W., 2016. Optimal stomatal behavior with competition for water and risk of hydraulic impairment. *Proceedings of the National Academy of Sciences* 113, E7222–E7230. <https://doi.org/10.1073/pnas.1615144113>

#281: increased assimilation may also i) be routed to reproductive organs; ii) be stored; iii) be used to change tissue allocation.

Response: Thank you for the suggestions, we have revised this sentence as:

“as leaf biomass increase is limited by light and nutrient availability, sink strength (i.e., the capacity of enzymes to assimilate carbon), as well as the capability of plants to use and allocate excess carbon into other organs^{39,44}”

#282: eCO₂ only leads to water saving if plants optimize for water use efficiency. This is not true in many ecosystems in which competition for water (and nutrients in the water) are important – in those systems if one plant does not transpire this water is consumed by other plant.

Response: Thank you for this suggestion. We agree that when plants compete for water, their optimization goal may be different than maximizing carbon gain over water loss as predicted by individuals. We have added discussion on this issue. See our response to your comments on line 262. Here in this second paragraph of the discussion, we are discussing this topic under the context of non-dryland, where water competition is not severe. Multiple evidence has shown that eCO₂ will save water and lead to increased runoff in these regions (Gedney et al., 2006; Ainsworth & Long 2005).

Ainsworth, E.A., Long, S.P., 2005. What have we learned from 15 years of free-air CO₂ enrichment (FACE)? A meta-analytic review of the responses of photosynthesis, canopy properties and plant production to rising CO₂. *New Phytologist* 165, 351–372. <https://doi.org/10.1111/j.1469-8137.2004.01224.x>

Gedney, N., Cox, P.M., Betts, R.A., Boucher, O., Huntingford, C., Stott, P.A., 2006. Detection of a direct carbon dioxide effect in continental river runoff records. *Nature* 439, 835–838. <https://doi.org/10.1038/nature04504>

#332-334: how does the model behave after removing datapoints if it is temporally dependent (t and t-1 in equation 2)?

Response: Part of our study area may be temporally covered by snow. It is important to remove these snow-covered pixels so that the NDVI can reflect valid information on plant canopy changes. Removing these pixels (mostly three or four consecutive months in the non-growing season) does not affect the model calculation since these snow-covered periods will be replaced with NA and the length of the time-series remain the same.

#362: as the coefficients are not in the equation, maybe mark it as “function of” (f(dNDVI) + f(dPREC)) to be correct. I miss a note on the format of the error term.

Response: Thank you for the suggestion, we have revised the equation and add the error term.

$$\delta NDVI_t = \theta_{NDVI} \delta NDVI_{t-1} + \theta_{prec} \delta Prec_{t-1} + \theta_{temp} \delta Temp_t + \theta_{cloud} \delta Cloud_t + \varepsilon$$

We made similar adjustment to the Eq. (3) as well.

#361: what does the delta stands for? Is it just to represent the deseasonalized+detrended? If so, note it in the text.

Response: Yes, it means de-seasonalized and detrended anomaly. We explained this notation in the revised manuscript (L352).

#364: anomaly is for cloud fraction only, not for precipitation (anomaly)? Anomaly is the difference from the whole time series mean?

Response: All variables used in the model area de-seasonalized detrended anomaly. The trend and mean seasonal cycle are calculated from the whole time series. We have rephrased this sentence to:

“where $\delta NDVI_t$, $\delta Prec_{t-1}$, $\delta Temp_t$ and $\delta Cloud_t$ represent anomalies for NDVI, precipitation, temperature, and cloud fraction, respectively.”

#384: was the pixel (grid cell) size noted somewhere?

Response: All datasets are in $0.5^\circ \times 0.5^\circ$ spatial resolution. We added description of this in the dataset section.

#384: should you not use the cloud fraction of the preceding month? For sure the NDVI state in any given time is a function of the previous, and not immediate, light availability.

Response: NDVI is strongly related to the fraction of light absorption by the green vegetation. Its value, together with concurrent climate (especially radiation), directly determines the gross primary production (GPP). The photosynthetic carbon fixation from current month is directly used for canopy leaf growth and many other physiological processes. Considering satellite NDVI for a given month is an average for multiple observations within the month, it should represent both the photosynthetic capacity (related to the amount of green leaves) and the growth of the leaf (allocation of newly fixed carbon). Therefore, we consider cloud cover for the current month, which directly affects the current month GPP, and resultant carbon accumulation in leaves as the factor in our model.

To test this, we build two models that uses either use current month temperature and cloud cover, or previous month temperature and cloud cover, with the same autocorrelation term and precipitation term. The model that uses current month climate data show better performance for more than 78.8% of the surface area (Fig. R7). In addition, considering cloud cover decrease solar radiation, and should have a negative effect on GPP and vegetation canopy growth, only the model that uses current month climate variables exhibit this negative relationship for most study area.

Figure R7. Comparison between the performance of the model that uses previous month climate variables and model that uses current month climate variables. (a) difference between the model performance (previous-current). (b) coefficient for the cloud fraction for the current month model. (c) coefficient for the cloud fraction for the previous month fraction.

#384: the DLM model is rather complex. I wonder how predictor independency may affect the results. Do precipitation anomaly and cloud fraction anomaly (I suppose they should)? If yes, how do you separate the effects.

Response: Thank you for raising this issue. We agree that, like multivariate linear regression, the independency of the input variable directly affects the DLM performance and effectiveness in separating the respective contribution. However, considering that all input and target variables are de-seasonalized and detrended, the correlation between each two of them is relatively weak. Since the precipitation is from the previous month and the cloud fraction is for the current month, the correlation between them is even weaker (Fig. R8).

We also tested a univariate DLM which only consider lag-1 auto correlation and pre-month precipitation, the results are very similar to the one we show in the main text. This suggests the effect of collinearity does not affect our conclusion.

R between cloud and previous month precipitation

Figure R8. Correlation between de-seasonalized detrended anomalies of cloud cover and previous month precipitation.

#384-387: so you fitted equation 2 using a DLM?

Response: Yes and no, the DLM essentially predicts current timestep NDVI anomaly using previous month NDVI anomaly, and previous month precipitation anomaly, current month temperature anomaly and current month cloud cover anomaly. All anomalies mentioned above are de-seasonalized and detrended. However, we do not consider estimating the coefficient for each variable as a “fit”, instead, we use a Kalman Filtering process to get the posterior estimate of the coefficient, which is based on prior information from the previous steps and the observation for the current step.

#417-419: it is not immediate clear why you divide in two. You can be more clear you are comparing the sensitivity from period 1 with sensitivity from period 2. I like this analysis, it is simple and conservative, thus robust; the average reader will understand what was done without problem, while the DLM is not trivial but probably much more sensible.

Response: Thank you for the suggestion, we have modified the text as below.

“The entire study period was split into two halves, i.e., from 1981 to 1998 and from 1999 to 2015, and we compared the sensitivity changes between the two periods. We use multivariate linear regressions to estimate the sensitivity for precipitation for both periods.”

We agree that the multivariate linear regression is simpler and easier to understand. But DLM allows us to better understand the dynamic changes of the sensitivity, whether the trend is robust or not, etc. We therefore keep the DLM as the main results.

#443 MLR – this only appear two time in the text, no need for acronyms.

Response: We have removed this acronym.

#443 – why here you calculate by decade and above you divide the period in two? Maybe keep the same approach in both of them?

Response: Thank you for your suggestion, here we use this regression for each decade because models provide much longer period of observations. By analyzing the precipitation sensitivity for each decade, it allows us to flexibly calculate the trend for longer or shorter periods. The results

are very consistent if we use a longer period to calculate the θ_{prec} trend. Additionally, although both methods show very consistent estimate of the trend, the regression method actually provides more robust estimate of the trend than using the difference between the two periods (Fig. R9).

Figure R9. A comparison between the trend in θ_{prec} estimated from the difference between two periods (x-axis) and regression from multiple decades (y-axis). Each point indicates the median value of relative θ_{prec} trend (trend of θ_{prec} divided by mean of θ_{prec}) for either dryland or non-dryland for each model-scenario combination.

#445 – what do you mean by normalized? I imagine it was a scaling or centralization and not a normalization process. Did you used a z-score approach? Would it not be better to say you compared the variances after scaling centralizing?

Response: We divided the trend by the mean value so that it can be regarded as percentage change per year. After this normalization, the results from LAI and NDVI can be directly compared. We had revised the statement to clarify this.

“the trend of θ_{prec} calculated from LAI is normalized by the mean θ_{prec} to get a relative trend for each pixel.”

Reviewer #3 (Remarks to the Author):

Comments to NCOMMS-21-49872

Zhang et al ‘s work tried to present changes in dryland ecosystems sensitivity to precipitation and explore the potential mechanism behind. While many similar studies have been done in this field, Zhang’s work further advanced our understanding of the underlying mechanism of the rising differences of ecosystem response to climate change. The work was done nicely, insightfully and

written quite well. However, this manuscript is very long and is easy to get me lost, I would suggest to shorten or discard some general descriptions, such as L206-216 and L416-429. Below is a subset of my concerns and note that the authors may flexibly modify some comments just in case I'm wrong at some place.

Response: We appreciate the reviewer's positive comments on our research. We agree that the current manuscript has very dense material, and some presentations are not very necessary and may distract the readers' attention. We have made substantial changes following the reviewer's suggestions. Please refer to our detailed responses below.

Title

The analysis actually have included both wet regions and drylands, and further, I am not sure if it is so correct to use dryland ecosystem in the title as only vegetation greenness are studied and maybe this could be more specific, like drylands vegetation greenness.

Response: We appreciate the reviewer's suggestion. In the title we focused on the aspect of analysis which we believe is more interesting to most readers. We like the title you suggested, and we have followed your suggestion and revised the title. It now reads: "Increasing sensitivity of dryland vegetation greenness to precipitation due to rising atmospheric CO₂"

Abstract

L26-28 I think the research gap introduced here is not so correct since many similar studies have been done (e.g., referring to relevant references). My question is that could we say the sensitivity of precipitation determine global /regional vegetation dynamic, as precipitation is not always the necessary driver of vegetation growth, like tropical regions. This also raise another issue, the sensitivity analysis in wet regions, does the change in sensitivity make sense? It could be also related to the equation (2), how much precipitation contribute to vegetation variations when considering NDVI at time of lag-1.

Response: We agree that a number of studies have investigated the variation of vegetation productivity or greenness to precipitation. However, as far as we know, very few focus on the response of this sensitivity to global climate change. Here, we highlighted this knowledge gap.

We understand the reviewer's concern on the relationship between precipitation sensitivity and the vegetation dynamics. The precipitation sensitivity may be a calculated "apparent metric" rather than an intrinsic property of the ecosystem. We agree that tropical ecosystems are mostly not water limiting, but precipitation still play an important role in regulating the seasonal and interannual variations of the canopy (Hilker et al. 2014, Guan et al. 2015, Jiang et al. 2019) and the intrinsic water use efficiency (Adams et al., 2019). This suggest that even in the tropical forest, a vegetation sensitivity to precipitation is still meaningful. We motioned this in the introduction.

In our analysis, the lag-1 autocorrelation is always considered. If not, the calculated precipitation sensitivity would be greater. The trend in autocorrelation is not likely to explain the trend of precipitation sensitivity. Here we show the correlation between the two is rather weak (Fig. R10).

Nevertheless, we agree with the reviewer that the sentence is not accurate, we revised it as:

“The sensitivity of vegetation growth to precipitation strongly regulates global and regional vegetation dynamics and their responses to drought”

Figure R10. Correlation between the trend of autocorrelation coefficient and the trend of precipitation sensitivity.

Adams, M.A., Buckley, T.N., Turnbull, T.L., 2019. Rainfall drives variation in rates of change in intrinsic water use efficiency of tropical forests. *Nature Communications* 10. <https://doi.org/10.1038/s41467-019-11679-8>

Guan, K., Pan, M., Li, H., Wolf, A., Wu, J., Medvigy, D., Caylor, K.K., Sheffield, J., Wood, E.F., Malhi, Y., Liang, M., Kimball, J.S., Saleska, S.R., Berry, J., Joiner, J., Lyapustin, A.I., 2015. Photosynthetic seasonality of global tropical forests constrained by hydroclimate. *Nature Geoscience* 8, 284–289. <https://doi.org/10.1038/ngeo2382>

Hilker, T., Lyapustin, A.I., Tucker, C.J., Hall, F.G., Myneni, R.B., Wang, Y., Bi, J., Mendes de Moura, Y., Sellers, P.J., 2014. Vegetation dynamics and rainfall sensitivity of the Amazon. *Proceedings of the National Academy of Sciences* 111, 16041–16046. <https://doi.org/10.1073/pnas.1404870111>

Jiang, Y., Zhou, L., Tucker, C.J., Raghavendra, A., Hua, W., Liu, Y.Y., Joiner, J., 2019. Widespread increase of boreal summer dry season length over the Congo rainforest. *Nature Climate Change* 1. <https://doi.org/10.1038/s41558-019-0512-y>

L33 Specify the models used, statistically or physically

Response: We have specified that these are terrestrial biosphere models in the revised manuscript.

L34-38, I like this explanations, they sound very convincing, maybe switch the description of wet and dry regions since the drylands was introduced firstly above.

Response: We appreciate the reviewer’s suggestion, but after careful consideration, we decide to keep it as is, since this is a more intuitive way for readers to understand the mechanism.

L58 I think reference 18 doesn't make sense.

Response: We have removed this reference and tree height, instead, we add rooting depth as another ecosystem characteristic that affects the precipitation sensitivity.

L73 Specify the data set and period studied.

Response: We specified the remote sensing dataset being used and the study period in the revised manuscript.

74-75 I think you do not remove dry season, actually GIMMS data in dry season have large uncertainty.

Response: We did not remove the dry season observations from the analysis. Satellite NDVI is known to be affected by cloud and aerosols. We use quality filtering layers and spatial aggregation to reduce such effect. In the dry season, there should be less cloud and aerosol, and the data quality is expected to be better. Will the reviewer provide any reference for this claim?

We actually separated the dry season and wet season and calculated the sensitivity trend for both seasons separately (Fig. S9). The results show the sensitivity estimated during dry season has large variation and this may be due to the small variation of the precipitation. However, the dynamic linear model is very robust in handling these noises and predict very consistent results.

L78 I don't think coefficient are calculated at each time step is an appealing strength.

Response: Since this study focuses on the trend of the vegetation sensitivity to precipitation, obtaining a robust time-varying estimate of this sensitivity is quite important. DLM allows us to calculate this sensitivity at each time-step with other factors and autocorrelation properly considered. We revised this sentence to better explain this.

“The approach is based on a dynamic linear model (DLM), which can estimate the time-varying relationship between environmental factors and NDVI from GIMMS (Methods), allowing us to derive a robust estimate of precipitation sensitivity with autocorrelation and other climate factors properly considered.”

L94 and Fig 1 should be better to add a histogram as inset to indicate how many pixels showing a decrease or increase in sensitivity. It's not so prone to catch some obvious pattern due to scattered pixels.

Response: We appreciate the reviewer's suggestion. We have added a probability density plot of the trend of precipitation sensitivity for the dryland and non-dryland (new Fig. 1 or Fig R11).

Figure R11. Same as Figure 1, but with subplot showing the probability density plot.

L118-119 the sensitivity trends were binned by aridity, right? If so, how big is each bin?

Response: The bin is based on aridity, we used 10 bins in here, with each equally-sized bin containing 3172 samples.

L122-123 I think this sentence should be rephrased; increased variability of vegetation greenness is caused by increased precipitation variability if an increase in sensitivity is demonstrated. I actually get lost when reading this part about variability, I just feel it doesn't contribute much to the results.

Response: Thank you for the suggestion. Here we mean that since we find an increase in sensitivity, this may lead to an increased variability in vegetation greenness, especially for drylands. This can be interpreted as a consequence of the change of precipitation sensitivity. This sentence now reads:

“The increase of θ_{prec} in drylands suggests a potentially increase of vegetation greenness variability.”

L206-216 better to move this paragraph to method

Response: We appreciate the reviewer's suggestion. But we respectfully disagree since this equation together with the results in Supplementary Fig. S16 help us understand how CO₂ affects these major components that contribute to the precipitation sensitivity changes.

We nevertheless followed the reviewer's suggestion and moved part of the description to the method section so that the readers can focus on the key results.

L218 A convincing explanation and this analysis is insightful.

Response: Thank you!

L262 relationship?

Response: Here we mean the relationship between plant and water, since our study focus on the plants' sensitivity to precipitation. We have revised this to "plant-water relationship".

L317 How do authors think about that the sensitivity is expected to decrease due to more frequent and more server climate extremes, it should be right?

Response: In this study we demonstrate that CO₂ leads to higher sensitivity vegetation greenness to precipitation. Here we mean that considering the precipitation variability is expected to increase as predicted by climate models, the impact on vegetation greenness is expected to be larger in dryland.

We tested whether there is any linkage between the trend in precipitation sensitivity and the trend in precipitation variability (Fig. R12). The results suggest very weak correlation between them for both drylands and non-drylands. We therefore don't expect that the sensitivity would decrease with more climate extremes.

Figure R12. A comparison between trend in precipitation sensitivity (x-axis) and trend in precipitation variability (y-axis) for dryland (a) and non-dryland (b).

Method

GIMMS data during dry season has poor quality and the temporal coverage should be updated to present.

Response: Please refer our response to your comments on Line 74-75 regarding to the data quality issue. We used the most recent version of GIMMS 3g v1 in our analysis, which only covers 1981-2015. We cannot update the temporal coverage to current due to this data limitation, but considering this dataset provide 35 years of observation, we believe the signal we observed is robust.

I just feel that NDVI at previous month could contribute a very large proportion to present NDVI? Do you test it?

Response: The previous month NDVI anomaly (de-seasonalized detrended, and hereafter) is used to predict the contribution of autocorrelation to the current month NDVI anomaly. The idea is that the anomaly from previous month will propagate to current month if no external forcing (environmental factors) is applied to the system. The sensitivity to external forcing will be better characterized with this autocorrelation being considered (Seddon et al., 2016). We calculated the average contribution of previous month NDVI anomaly (Fig. R13), the results suggest it contribute 10.3% of current month NDVI anomaly variation.

Figure R13. The spatial pattern of variance of NDVI anomaly explained by the previous month NDVI anomaly.

Seddon, A.W.R., Macias-Fauria, M., Long, P.R., Benz, D., Willis, K.J., 2016. Sensitivity of global terrestrial ecosystems to climate variability. *Nature* 531, 229–232. <https://doi.org/10.1038/nature16986>

Do we need to introduce Section 3-4 in such detailed? I think few readers are interested in reading them.

Response: Thank you for your suggestions. The methods section is indeed very long and difficult to follow. We agree the detailed information of DLM and multivariate regression are not appealing to most readers. We therefore moved them to the supplementary information, and briefly introduce the two methods in the “Precipitation sensitivity of vegetation” section.

I like this analysis in L453, but I am not sure what’s basis for the separations of LAI/T, T/ET, ET/P

Response: Here we are trying to understand the what factors drive the LAI sensitivity to precipitation changes. Considering water is the used by plants through transpiration (there is a tight relationship between LAI and T), and T is part of ET, the ratio between two is an important ecohydrological metric, and the ET to precipitation fraction is also well studied in hydrology for decades. These three factors provide a complete chain how water is used by plants, with each factor has its own meaning. This is also important for us to understand how CO₂ affects the precipitation sensitivity through different ecohydrological processes. We added additional explanation for this in the revised manuscript.

“ θ_{prec} can be approximated by LAI sensitivity to precipitation and further decomposed to three components with well-defined ecohydrological meaning.”

I cannot follow from L458 onward, maybe the other reviewers could make suggestions.

Response: We have thoroughly revised the manuscript and improved the clarity and logical flow. Specifically, we added a paragraph at the beginning of the section and briefly summarized the goal of this section, including what is the minimalistic model, why do we use it, and how does it work. We hope through these revisions, this section is easier to follow and can provide new insights into the CO₂ effect on plant water relationship.

Reviewer comments, second round -

Reviewer #1 (Remarks to the Author):

This is my second time reviewing this manuscript. Unfortunately, I feel I have to report that in my opinion, this paper just isn't written clearly and convincingly. I applaud the authors for a lot of hard work addressing an important issue (drivers of vegetation sensitivity to precipitation), but the way in which their results and methods are presented are still quite difficult to follow, and thus the submitted study is not as convincing as I feel it needs to be. I found myself having to re-read many sections multiple times to follow (I think) what the authors have done, and then at times I feel like I had to guess at what was implied. This manuscript reads like a mis-mash between three different papers, each of which has its own quite involved methods and results. Furthermore, it is not a synthesis of these methods, instead they are described in series. Often, the way in which the 'sensitivities' and 'variabilities' of one variable in response to another as driven/conditioned by other variables (e.g. variability in NDVI in time as driven by precipitation variation) is just presented in a way I find really challenging to interpret. The amount of supplementary figures (20!), tables, and text, is a quite large. That the authors feel that so much supplementary material is needed suggests that the points are not being made clearly (or that too many are being made and diluting your message). For all these reasons maybe a more technical journal with more space is needed to convey this story.

Reviewer #3 (Remarks to the Author):

The authors have carefully reviewed my comments and provided appropriate responses to them. The new information and changes added to the manuscript has greatly improved it. I believe the manuscript is appropriate for publication now.

Reviewer #4 (Remarks to the Author):

The authors investigated the precipitation sensitivity of vegetation greenness (as sensed remotely with NDVI). Aggregating Earth's vegetated surface into two categories—dryland, and not-dryland, using an aridity index, the authors analysis revealed that while dryland greening is becoming more sensitive to precipitation, the opposite is true for not-drylands. This is a significant and important finding. This revised manuscript reads well, and was not as difficult to follow as prior iterations seem to have been. I was not an initial reviewer for the manuscript, but have read the prior reviews and the authors' response.

Regarding the response to prior reviewers' comments, the authors have replied to each comment carefully, almost always including revision of the manuscript and/or additional analyses and supplementary material as a result. Thus, the revision does seem thorough and for the most part addresses the concerns of prior reviewers. I share the concern about interpreting the aridity index primarily through the lens of precipitation, and the authors should at the very least discuss the influence of precipitation, compared to potential evapotranspiration (PET), on the range of observed AI values. In their own response to prior review, the authors state regarding their observed increased dryland precip. sensitivity, decreased non-dryland precip. Sensitivity, ..."this suggest water availability may not be driven by precipitation change, but other factors." Given the delineation of the two major contrasts throughout this paper (drylands, non-drylands) was an aridity index with two terms—P and PET, the lack of exploration of the components of PET (most importantly, air temperature, but also insolation/wind) represents a missed opportunity for this study.

The authors have not yet fully caveated the oversimplification of placing Earth's diverse vegetated systems into two bins—dryland or not. Response to prior reviewers resulted in acknowledging that precipitation change alone cannot explain the contrasting precip. sensitivity trends, but the authors have not yet added in (as suggested by prior reviewer, and I concur it still needs to be

done) to the discussion some text to caveat the findings. I suggest this can be done at the end of the second paragraph of the discussion, which already calls attention to further factors which may be influencing the results. There, the authors could also mention the components of PET, especially air temperature, which likely influenced the aggregation of Earth's various systems into dryland or not, according to the AI method. For example, in Figure R3, the authors show other climate variables (including temperature trend, and cloud cover trend besides precipitation trend), but here it would be most appropriate to include PET trend.

To further agree with prior reviewers' concerns regarding the simplicity of the Aridity Index method, while I agree that the balance of 'supply and demand' is a good starting place, the distribution of rainfall annually may leave many periods where plants are under severe drought stress (e.g., Mediterranean climates), sometimes going more than half the year without rain, while the 'mean annual precipitation' is concentrated in a narrow portion of the year. In this way, for a single location the AI would classify it at the monthly timescale as a dryland for some months, and not for others. I think this is at the heart of the concern with only including Aridity Index. Prior reviewers suggested also including a version of the analysis that partitioned the finding by biomes—even if a coarse framework (e.g., Whittaker biomes) was used, this could be a helpful form of analysis and reveal where the divergent precipitation sensitivity holds based on biome-binned AI categories of dryland/not-dryland, and where it does not. While this may result in some biomes not having significant data for one or the other AI-based bins, many of them would, and any difference in sensitivity of greenness to precipitation would be interesting, and likely provide a needed roadmap for future work.

Line-specific comments: (line #'s from the version with tracked changes expanded)

Line 79: Add a parenthetical with some examples after 'environmental factors' (e.g., example 1, example 2). A few specific examples will help guide the reader to what you mean.

Line 80: add "remotely sensed" before "from GIMMS"

Lines 94-95: Here you introduce AI, but do so briefly. The CGIAR dataset from which it was taken, shown in figure S1—was this calculated over some long-term period (30 years)? Is AI considered fixed for the entire study? Are areas once a dryland always a dryland, and once a non-dryland always a non-dryland? These were things that had me confused from early on while reading the present version of the manuscript, so some additional text here to explain will be helpful. I do think that in removing many technical details previously the manuscript now seems quite readable, but some detail on this very central decision—to aggregate Earth's vegetated systems into one of two categories—deserve some careful detail when introduced. Understanding how this was done and why, will prevent later confusion.

Line 109: add "greening" in two places: after drylands, and after non-drylands. We cannot say if overall the drylands are becoming more sensitive to precipitation—but their greening (as sensed via NDVI) is.

Line 179: rooting moisture is definitely important, and this will also lead to longer periods of maximum stomatal closure—when the concentration of CO₂ could be 200 or 1000, without much influence on plant water use. Some acknowledgement that stomata are not always open, especially in drylands, is important here. Stomatal operation may be limited for weeks to months in some dryland systems, especially with uneven seasonal distributions of precipitation and soil moisture.

Line 194: Specifically, you've tested mean annual temperature and cloud cover—what about PET? As the other component of the AI used to bin ecosystems, it would be an important one to test and add to this new supplemental figure.

Line 201: "climate change" is too broad a statement as an example of a factor—can you be more specific here? Warming? Increased PET? Changing distributions of P? Are a few of a long list of potential examples that could be read into "climate change".

Line 219: (Fig. 3) – this figure shows that CO₂ effects are small in drylands relative to non-drylands, as one would expect in places that stomata are less often open.

Line 233: add "in non-drylands" after CO₂, see comment on Fig. 3 above.

Line 266-267: I'm unsure how 'partial' the stomatal closure is. Does this mean part of the year, or that stomata apertures are not fully open, or what? It's a complex phenomena at the scale of a single leaf, let alone at the global scale. I could not follow if the model accounted for some continuum of potential stomatal operation, or what. If stomatal closure is to be invoked, please see my general comments (and line-specific ones above) that plants with closed stomata under drought conditions don't realize a WUE benefit.

Line 278-280: C3 vs C4 is an important distinction, I agree. Also, plants have a wide range of stomatal behaviors. Sometimes these are aggregated at two scales (as with the present paper's ecosystem binning), but practice, plants have a wide continuum of stomatal responses—close early during water deficit, close late—and these are often tied to other important points raised by other reviewers (competition for water, in water-limited systems). This would be a good place to add 1-2 sentences acknowledging this complexity that is not (and cannot yet be) accounted for given the present state of models.

Line 272: The third effect is the same as the first? I do not see the distinction.

Line 311-313: When stomata close, leaves still lose water. A higher LAI in a dryland would still lead to amplified water loss, mechanistically, since there is more surface area across which the plant's minimum conductance will shunt water to the atmosphere.

Line 333: Another good place to add "stomatal sensitivity to water deficit/drought" as another factor to consider in further/future study.

Reviewer #5 (Remarks to the Author):

Summary: Zhang et al. use a combination of remotely sensed observations (NDVI and LAI estimates), global climate data, and semi-factorial model simulations of LAI (from MsTMIP) to examine the sensitivity of vegetation to precipitation, and more specifically how the sensitivity to precipitation has changed globally in both dryland and non-dryland systems. They find that the sensitivity of vegetation greenness/LAI to precipitation has increased in drylands and decreased elsewhere. They also find that these changes in vegetation sensitivity to precipitation are mostly driven by CO₂ fertilization, with increased LAI in drylands driving their enhanced sensitivity and decreased stomatal conductance in non-drylands driving their reduced sensitivity. Overall, the study seems well designed and well executed, and the authors have performed a number of sensitivity tests to ensure that their results are robust to methodological choices and assumptions. While much previous research has shown that vegetation in dry regions is more sensitive to precipitation than in wet regions, this work makes a significant and novel contribution by showing that these sensitivities have changed across much of the world (and vary by aridity) and that this is largely driven by CO₂ effects. I think the authors have mostly done a good job of responding to previous reviewer comments, and I do not have any major concerns about the work, mostly just some clarifying questions and suggestions on presentation.

General comments:

1) I think there's way more detail about the ecohydrological model in section 4 of the methods than is actually needed in the main text, and I would suggest moving much of it to the supplement. (Note: I don't have the relevant technical expertise to comment much on the quality/appropriateness of the ecohydrology model itself, so I'll leave that to other reviewers.) Eqn 6 seems useful to have in the main Methods section, but to me, Eqns. 7-14 seem like a little more detail than necessary for the main text. It might be easier for readers to follow and understand to have descriptions of the main logic of the model (and the sensitivities being derived from it) and save the detailed description/equations for the supplementary text, similar to what the authors did in response to the first round of reviewer comments on the DLM and MLR methods.

2) I agree with reviewer #1 (their comment on L167) that the aggregation into dryland and non-dryland at a global scale is washing out a lot of important regional variability. I understand that it's not possible in this kind of paper to delve too much into specifics of every region, but to me, it seems like a little more attention could be paid to whether and to what extent some regions deviate from the global aggregations.

Specific comments:

Lines 122-127: this seems to conflict with both theory (O'Gorman & Schneider 2009, Pendergrass et al. 2017) and observations (Georgi et al. 2011) of how precipitation variability changes in a warming climate. Why might this be?

Lines 179-182: Just to clarify, are these confidence intervals accounting for both inter-model differences *and* spatial variability within the regions?

Lines 186-194: Eqn. 1 is clearly mathematically valid, but the “well-defined ecohydrological meaning” of each term isn’t necessarily clear to me. Maybe a brief, clear explanation of those ecohydrological meanings would be helpful here?

Lines 221-222: What’s the mechanism by which CO₂ could change PET?

Lines 234-235: The meaning of the lines in 4b is not necessarily clear from the legend in the figure. I’d suggest writing clearer descriptions of their meaning in the caption.

Lines 274-275: By “low (dry) and high (wet)”, do you mean that the CO₂ effect in low vegetation regions leads to drying of streamflow (enhanced LAI outweighs reduced conductance) and vice versa? If so, I would suggest just being a little clearer because it wasn’t immediately clear to me what “low (dry) and high (wet)” meant.

Lines 292-305: I like this last paragraph a lot. Great point about “greening but drying,” and a nice way to finish the paper!

Lines 328-329: The performance of LAI_{3g} would depend not just on NDVI_{3g}, but also on the performance of the MODIS LAI product, correct? And in some regions the LAI estimates from MODIS can be a little suspect, I think?

Lines 408-409: Is this supposed to say “Since *not* all models participate...”? This sentence could also generally be better worded I think.

Line 448: I’m not a hydrologist by any means, but it seems like interception should vary by both the amount of LAI (more LAI = more interception?) and by the intensity of precipitation (light precipitation events should have a greater percentage intercepted than heavy precipitation events, since once canopy has reached a maximum interception capacity, any additional precipitation would not be intercepted?) Is this 15% interception a common practice?

Figure S14: Would it be worth analyzing the relationship of theta-prec to VPD and/or PET trends?

Figure S16: I really like this figure and would suggest possibly adding it as a panel in Fig. 4. Maybe instead of the soil texture analysis? To me, it seems like the soil texture analysis is more of a supplemental thing and not nearly as crucial to the main argument as Figure S16 is.

References:

- Giorgi, F., Im, E. S., Coppola, E., Diffenbaugh, N. S., Gao, X. J., Mariotti, L., & Shi, Y. (2011). Higher hydroclimatic intensity with global warming. *Journal of Climate*, 24(20), 5309–5324.
- O’Gorman, P. A., & Schneider, T. (2009). The physical basis for increases in precipitation extremes in simulations of 21st-century climate change. *Proceedings of the National Academy of Sciences*, 106(35), 14773–14777.
- Pendergrass, A. G., Knutti, R., Lehner, F., Deser, C., & Sanderson, B. M. (2017). Precipitation variability increases in a warmer climate. *Scientific Reports*, 7, 17966.

**Increasing sensitivity of dryland vegetation greenness to precipitation due to rising
atmospheric CO₂
NCOMMS-21-49872A
Response to Reviewers**

We appreciate the constructive comments from the reviewers and the invitation from the editor to submit a revised version. We have carefully followed the reviewers' suggestions to carry out additional analyses and improve our manuscript. Please see below our point-to-point responses in blue text following reviewer comments. All line numbers and figures numbers in this response letter refer to the clean version of the revised manuscript.

Reviewer #1 (Remarks to the Author):

This is my second time reviewing this manuscript. Unfortunately, I feel I have to report that in my opinion, this paper just isn't written clearly and convincingly. I applaud the authors for a lot of hard work addressing an important issue (drivers of vegetation sensitivity to precipitation), but the way in which their results and methods are presented are still quite difficult to follow, and thus the submitted study is not as convincing as I feel it needs to be. I found myself having to re-read many sections multiple times to follow (I think) what the authors have done, and then at times I feel like I had to guess at what was implied. This manuscript reads like a mis-mash between three different papers, each of which has its own quite involved methods and results. Furthermore, it is not a synthesis of these methods, instead they are described in series. Often, the way in which the 'sensitivities' and 'variabilities' of one variable in response to another as driven/conditioned by other variables (e.g. variability in NDVI in time as driven by precipitation variation) is just presented in a way I find really challenging to interpret. The amount of supplementary figures (20!), tables, and text, is a quite large. That the authors feel that so much supplementary material is needed suggests that the points are not being made clearly (or that too many are being made and diluting your message). For all these reasons maybe a more technical journal with more space is needed to convey this story.

Response: We appreciate the reviewer's time and efforts in reviewing this manuscript. As the reviewer suggests, the manuscript focus on an important issue, i.e., the drivers of the vegetation sensitivity to precipitation. We appreciate that the reviewer found the logic flow of the manuscript to be clear and straightforward. In our analysis, we first found contrasting trends of the vegetation sensitivity to precipitation in global dryland and non-dryland regions using satellite observations; to understand what caused these contrasting trends, we used both terrestrial biosphere models and a simple minimalistic model and revealed that the different responses of vegetation to CO₂ in dryland and non-dryland is the major cause. The multiple novel methods served the same goal for the characterization and explanation of the trends in vegetation sensitivity to precipitation, and should not be separated papers.

We acknowledge, however, that the material presented is dense at times, particularly given the multiple methods, datasets, model outputs and statistical methods we employed. We used such a broad array of approaches and data to ensure that the reported phenomenon is robust, and the revealed mechanism advances our understanding of the climate change impact on terrestrial

ecosystem. We believe such rigor is required by the high standard of Nature Portfolio journals. These additional robustness tests also add to the number of figures in the supplementary information, but we feel these are necessary and will be interesting for those readers who want to test other datasets or methods. Ignoring these additional supplementary tests does not directly affect the interpretation of the main message in the main text.

We have endeavored to improve the accessibility of the writing and reduce the density of the text throughout the review process, and each round of reviews has led to significant improvements in this regard (as noted in this most recent round by the new reviewer #4, who found the manuscript quite readable), including additions to the supplementary material. We feel however that there is a limit to the degree to which we can or should simplify our analysis. We prefer to retain the robustness analyses and the multiple data sources, rather than excluding them to simplify, as we feel that although it makes it a challenging paper to review, they will make for more compelling results for the reader once published. We hope that the reviewer and editor understand this choice and we can agree to disagree on the ideal degree of complexity such an analysis should entail. We also hope that the improved writing helps assuage the reviewer's concerns.

Reviewer #3 (Remarks to the Author):

The authors have carefully reviewed my comments and provided appropriate responses to them. The new information and changes added to the manuscript has greatly improved it. I believe the manuscript is appropriate for publication now.

Response: Thank you for your efforts for evaluating the manuscript. We are pleased to see that you are satisfied with our revisions.

Reviewer #4 (Remarks to the Author):

The authors investigated the precipitation sensitivity of vegetation greenness (as sensed remotely with NDVI). Aggregating Earth's vegetated surface into two categories—dryland, and not-dryland, using an aridity index, the authors analysis revealed that while dryland greening is becoming more sensitive to precipitation, the opposite is true for not-drylands. This is a significant and important finding. This revised manuscript reads well, and was not as difficult to follow as prior iterations seem to have been. I was not an initial reviewer for the manuscript, but have read the prior reviews and the authors' response.

Response: Thank you for the positive comments on our manuscript.

Regarding the response to prior reviewers' comments, the authors have replied to each comment carefully, almost always including revision of the manuscript and/or additional analyses and supplementary material as a result. Thus, the revision does seem thorough and for the most part addresses the concerns of prior reviewers. I share the concern about interpreting the aridity index primarily through the lens of precipitation, and the authors should at the very least discuss

the influence of precipitation, compared to potential evapotranspiration (PET), on the range of observed AI values. In their own response to prior review, the authors state regarding their observed increased dryland precip. sensitivity, decreased non-dryland precip. Sensitivity, ...”this suggest water availability may not be driven by precipitation change, but other factors.” Given the delineation of the two major contrasts throughout this paper (drylands, non-drylands) was an aridity index with two terms—P and PET, the lack of exploration of the components of PET (most importantly, air temperature, but also insolation/wind) represents a missed opportunity for this study.

Response: We appreciate the insights from the reviewer. We agree that both precipitation and PET changes are important to the changes of aridity. Considering θ_{prec} is almost monotonically decreasing along aridity index (Fig. 1c), a contrasting trend of θ_{prec} in drylands and non-drylands may indicate that the aridity index has different trends for drylands and non-drylands. This is more likely to be caused by changes in precipitation rather than PET, considering PET is likely to increase with global warming. Actually, in our response to the first-round reviews, we also tested the effect of PET trend on θ_{prec} in the revised Figure S13. We did not find an obvious relationship between θ_{prec} and PET trend.

PET can be calculated from multiple different equations, e.g., Priest Taylor, Penman Monteith and its variations (Yang et al., 2019) and they can have large differences in the PET trend (Sheffield et al., 2012). Here we calculated the trend of two widely used PET datasets, i.e., the CRU TS 4.05 and the Princeton dataset. Although the time periods used to calculate the trends do not match exactly, their spatial patterns are different (Fig. R1). But most regions (72.8% and 88.8% for Princeton and CRU, respectively) show a positive trend and there is no obvious difference between dry and wet.

We further explored the relationship between θ_{prec} and PET trend and its component, for the entire study area and for dryland and non-dryland only (Fig. R2). The strongest correlation is found between θ_{prec} and precipitation, which is always negative for both dryland and non-dryland. θ_{prec} shows rather weak correlation with other factors, with correlation coefficient mostly below 0.1.

These analyses suggest that the contrasting trend of θ_{prec} between dryland and non-dryland is not caused by PET and its components. We added Figure R2 into the supplementary information and added discussions on this.

“Changes in potential evapotranspiration and its components (e.g., radiation, temperature) can also affect the water availability, but these factors show very weak relationships with θ_{prec} in terms of their trends (Supplementary Fig. S14).”

Fig. R1. A comparison between PET trend from Princeton and CRU. **a** PET trend estimated from Princeton Terrestrial Hydrological Research Group. The trend is estimated from 1980-2008. **b** PET trend estimated from CRU TS 4.05. The trend is estimated during 1980-2015. Note the large difference in terms of the magnitude.

Fig. R2. Relationship between trend of precipitation sensitivity and trend in climate variables (precipitation, temperature, cloud cover, potential evapotranspiration, and vapor pressure deficit). The first column shows the correlation for all pixels (a-f), second column for the dryland pixels (g-i), third column for the non-dryland pixels (m-r). PET from both Princeton (Sheffield et al., 2012) and CRU TS 4.05 (Harris et al., 2020) are used.

The authors have not yet fully caveated the oversimplification of placing Earth's diverse vegetated systems into two bins—dryland or not. Response to prior reviewers resulted in acknowledging that precipitation change alone cannot explain the contrasting precip. sensitivity trends, but the authors have not yet added in (as suggested by prior reviewer, and I concur it still needs to be done) to the discussion some text to caveat the findings. I suggest this can be done at the end of the second paragraph of the discussion, which already calls attention to further factors which may be influencing the results. There, the authors could also mention the components of PET, especially air temperature, which likely influenced the aggregation of Earth's various systems into dryland or not, according to the AI method. For example, in Figure R3, the authors show other climate variables (including temperature trend, and cloud cover trend besides precipitation trend), but here it would be most appropriate to include PET trend.

Response: We thank the reviewer for the suggestion. We agree that precipitation and potential evapotranspiration changes can be important for the vegetation sensitivity at local scale. However, at regional and larger spatial scales, these variations show limited effects since the local signals are smoothed out after aggregation. Specifically, we do not find strong relationship between θ_{prec} and PET and its components (Fig. R2). And as a result, these factors are not likely to explain the contrasting trend of θ_{prec} between dryland and non-dryland. We updated this Figure R3 (in the first-round review) and added it into the supplementary information as Figure S14. We also followed the reviewer's suggestions and discussed the potential effect of precipitation and potential evapotranspiration at local scale.

"It should also be noted that although trends in precipitation and potential evapotranspiration do not likely explain this contrasting trend of θ_{prec} at a global scale, they may play an important role in regulating the local water availability and thus contribute to the variation of θ_{prec} at local scale."

To further agree with prior reviewers' concerns regarding the simplicity of the Aridity Index method, while I agree that the balance of 'supply and demand' is a good starting place, the distribution of rainfall annually may leave many periods where plants are under severe drought stress (e.g., Mediterranean climates), sometimes going more than half the year without rain, while the 'mean annual precipitation' is concentrated in a narrow portion of the year. In this way, for a single location the AI would classify it at the monthly timescale as a dryland for some months, and not for others. I think this is at the heart of the concern with only including Aridity Index. Prior reviewers suggested also including a version of the analysis that partitioned the finding by biomes—even if a coarse framework (e.g., Whittaker biomes) was used, this could be a helpful form of analysis and reveal where the divergent precipitation sensitivity holds based on biome-binned AI categories of dryland/not-dryland, and where it does not. While this may result in some biomes not having significant data for one or the other AI-based bins, many of them would, and any difference in sensitivity of greenness to precipitation would be interesting, and likely provide a needed roadmap for future work.

Response: We appreciate the reviewer’s suggestions. The seasonal distribution of precipitation indeed can greatly affect the vegetation growth, especially in dryland when the precipitation is strongly skewed and mostly concentrated in the wet season (Nicholson, 2011). In our framework, the vegetation sensitivity to precipitation is evaluated at a monthly timescale, which yielded a robust sensitivity estimate. However, precipitation intermittency can still act as an independent axis on top of the aridity index.

Biome type distribution reflects the long-term adaption of the ecosystem to regional climate and may provide additional information. To this end, we evaluated the distribution of biome types along aridity index and how they would affect the trend of vegetation sensitivity to precipitation (Fig. R3). We found that biome distributions are strongly dependent of the ecosystem aridity (Fig. R3b), with forests in the wet regions and grasslands and shrublands in the dry regions. Importantly, 7 out of 8 biome types exhibit a decreasing trend of θ_{prec} along the aridity index. The only biome showing an increasing trend is tropical and subtropical dry broadleaf forest, which only occupies 3% of the study area. Collectively, they show a contrasting trend between drylands and non-drylands. However, it should be noted that considerable differences exist between biomes, which may be attributed to abiotic factors (e.g., precipitation trend, PET trend) and ecosystem characteristics. We feel this is important and we added this figure to the supplementary information and discussed it in the main text.

“The θ_{prec} trend within each biome shows a similar pattern along the aridity index as that from the entire study regions (Supplementary Fig. S15), suggesting that biome-specific characteristics are not the major cause for the contrasting trend.”

Fig. R3. Vegetation sensitivity to precipitation along aridity index for each biome type. **a** the Olson's biome map for our study region. **b** the distribution of biome along the aridity index. **c** relationship between θ_{prec} and aridity index within each biome type.

Line-specific comments: (line #'s from the version with tracked changes expanded)

Line 79: Add a parenthetical with some examples after 'environmental factors' (e.g., example 1, example 2). A few specific examples will help guide the reader to what you mean.

Response: Thank you for the suggestion, this sentence has been revised:

"....., which can estimate the time-varying relationship between environmental factors (e.g., precipitation, temperature, radiation) and NDVI from remotely sensed GIMMS (Methods),"

Line 80: add "remotely sensed" before "from GIMMS"

Response: revised as suggested.

Lines 94-95: Here you introduce AI, but do so briefly. The CGIAR dataset from which it was taken, shown in figure S1—was this calculated over some long-term period (30 years)? Is AI considered fixed for the entire study? Are areas once a dryland always a dryland, and once a non-dryland always a non-dryland? These were things that had me confused from early on while reading the present version of the manuscript, so some additional text here to explain will be helpful. I do think that in removing many technical details previously the manuscript now seems quite readable, but some detail on this very central decision—to aggregate Earth's vegetated systems into one of two categories—deserve some careful detail when introduced. Understanding how this was done and why, will prevent later confusion.

Response: The reviewer is correct, the CGIAR aridity index (AI) is calculated using the long-term mean annual precipitation and potential evapotranspiration during 1970-2000. AI is a general description of the long-term climatic water deficit so we did not consider a varying AI during the study period. We agree with the reviewer that this is an important issue so we highlighted that the aridity index is calculated from long-term mean precipitation over potential evapotranspiration in the revised main text and discussed the potential caveat in the discussion and methods section.

"Aridity index is an indicator of the degree of dryness calculated as the ratio between long-term precipitation and potential evapotranspiration²⁵. Here we use a static map of aridity index provided by CGAIR (Supplementary Fig. S1)."

"It should be noted that aridity index is a measure of long-term climatic dryness conditions and different potential evapotranspiration definitions and calculations can yield different potential evapotranspiration trends, we ignored aridity changes during the study period for simplicity."

We would also like to mention that our manuscript does not treat aridity as a dichotomy but a continuum from dry to wet. For example, in Figure 1, 4 and many others in supplementary information, the patterns along aridity index are presented. The contrasting trend from dry to wet shows a continuous change, but to attract the interest of the broad audiences, we highlighted the difference between dryland and non-dryland.

Line 109: add “greening” in two places: after drylands, and after non-drylands. We cannot say if overall the drylands are becoming more sensitive to precipitation—but their greening (as sensed via NDVI) is.

Response: Thank you for the suggestion. We revised this sentence as suggested, it now reads:

“..., suggesting that vegetation greenness in drylands becomes overall more sensitive to precipitation variations, while the greenness in non-drylands becomes less sensitive.”

Line 179: rooting moisture is definitely important, and this will also lead to longer periods of maximum stomatal closure—when the concentration of CO₂ could be 200 or 1000, without much influence on plant water use. Some acknowledgement that stomata are not always open, especially in drylands, is important here. Stomatal operation may be limited for weeks to months in some dryland systems, especially with uneven seasonal distributions of precipitation and soil moisture.

Response: We appreciate the reviewer’s insights. In most dryland ecosystems, precipitation has a strong seasonality and plants only grow during the wet season. During the dry season, perennial species reduce stomatal conductance to save water. There should be limited variations in both precipitation and vegetation greenness during the dry season, and thus little influence on the precipitation sensitivity trend. The DLM method is robust in handling these dormancy periods (Fig. S9).

In this paragraph, we described some observational evidence that some factors may also affect the changes (or trend) of precipitation sensitivity. We agree that plants’ seasonal dormancy (stomatal closure) is a widely-observed phenomenon in drylands, however, this does not provide direct linkage to the interannual variation of the precipitation sensitivity as we discussed in this paragraph. Additional descriptions will be needed if we want to fully explain the possible mechanisms, but this would also dilute the main idea here in this paragraph and decrease the readability.

Nevertheless, we agree with the reviewer that this point is noteworthy and we mentioned this in the discussion.

“Other factors including soil texture, rooting depth, water table depth, precipitation seasonality, and stomatal sensitivity to drought also affect ecosystem water availability and precipitation sensitivity, and their interactions with CO₂ needs to be further explored.”

Line 194: Specifically, you've tested mean annual temperature and cloud cover—what about PET? As the other component of the AI used to bin ecosystems, it would be an important one to test and add to this new supplemental figure.

Response: We have added two independent PET datasets into this comparison and updated the supplementary figure and associated discussions. Please see our response to your major comment #1.

Line 201: “climate change” is too broad a statement as an example of a factor—can you be more specific here? Warming? Increased PET? Changing distributions of P? Are a few of a long list of potential examples that could be read into “climate change”.

Response: In the MsTMIP model intercomparison protocol, climate refers to a number of climate forcings used to drive the terrestrial biosphere models, including incoming longwave/shortwave radiation, air temperature and humidity, pressure, wind speed, precipitation. These climate forcing are from the CRU-NCEP reanalysis data and can be regarded as actual change of the Earth climate in the past. We have added the climate forcing considered in the supplementary Table S1 where the MsTMIP simulation scenarios are described.

Line 219: (Fig. 3) – this figure shows that CO₂ effects are small in drylands relative to non-drylands, as one would expect in places that stomata are less often open.

Response: Here the figure shows the combined effect of CO₂ on vegetation sensitivity to precipitation. The relatively small effect of CO₂ is because multiple individual effects counteract with each other. We are not focusing on the absolute values here, but the difference between dryland and non-dryland to explain the contrasting trends between dry and wet. We have highlighted this in Line 174-175.

Line 233: add “in non-drylands” after CO₂, see comment on Fig. 3 above.

Response: We have revised this sentence to make it clear.

“Given the large effect of CO₂ in explaining the difference between drylands and non-drylands, we further quantify its impact on disentangled components controlling θ_{prec} by leveraging the model outputs.”

Line 266-267: I'm unsure how 'partial' the stomatal closure is. Does this mean part of the year, or that stomata apertures are not fully open, or what? It's a complex phenomena at the scale of a single leaf, let alone at the global scale. I could not follow if the model accounted for some continuum of potential stomatal operation, or what. If stomatal closure is to be invoked, please see my general comments (and line-specific ones above) that plants with closed stomata under drought conditions don't realize a WUE benefit.

Response: The ‘partial’ stomatal closure here refers to the reduced stomatal conductance in response to the increased ambient CO₂ concentration. This is predicted by all state-of-the-art stomatal conductance models and has been observed at ecosystem scale at flux tower and FACE experiment sites worldwide (Ainsworth and Long, 2005; Keenan et al., 2013). We agree with the reviewer that current presentation can be misleading, we therefore revised this sentence.

“The first effect is that eCO₂ can reduce the aperture of the stomata (reduction in stomatal conductance), so that the amount of water needed by plants decreases, and thus so does the sensitivity of transpiration to precipitation ($\frac{\partial E_T}{\partial P}$)”

This eCO₂ induced stomatal conductance reduction is different from the drought-induced stomatal closure, the latter depends on the drought severity and ecosystem-specific traits. And as the reviewer suggests, drought-induced stomatal closure does not always increase WUE. However, the eCO₂ effect on stomatal conductance is universal and will increase WUE (Keenan et al., 2013).

Line 278-280: C3 vs C4 is an important distinction, I agree. Also, plants have a wide range of stomatal behaviors. Sometimes these are aggregated at two scales (as with the present paper’s ecosystem binning), but practice, plants have a wide continuum of stomatal responses—close early during water deficit, close late—and these are often tied to other important points raised by other reviewers (competition for water, in water-limited systems). This would be a good place to add 1-2 sentences acknowledging this complexity that is not (and cannot yet be) accounted for given the present state of models.

Response: Thank you for the suggestion! We agree that the stomatal behavior, plant hydraulic traits, inter-specific competition are factors that directly affect the vegetation sensitivity to precipitation sensitivity. We followed the reviewer’s suggestion and added the following discussions to highlight these mechanisms are not considered in the current model and may lead to some uncertainty.

“It is worth noting that this simple model does not account for the variations in plant hydraulic traits and inter-species competition for water, which may both affect the vegetation sensitivity to precipitation^{18,34}, but their effects on the trend of sensitivity is still uncertain.”

Line 272: The third effect is the same as the first? I do not see the distinction.

Response: The first and third effect are the two sides of a coin. They are both caused by the CO₂ induced decline in stomatal conductance, but the first effect is on the sensitivity of transpiration to precipitation ($\frac{\partial E_T}{\partial P}$), the third effect is on the LAI sensitivity to transpiration ($\frac{\partial LAI}{\partial E_T}$). We did not mention these two effects together because both the first and second effects are on the transpiration sensitivity to precipitation, while the third is on the LAI sensitivity to transpiration. We have revised this sentence as below:

“The third effect is also due to the eCO₂-induced stomatal conductance reduction, which on the other hand, reduced transpiration per leaf area. This is equivalent to a universal increase of water use efficiency and $\frac{\partial LAI}{\partial E_T}$.”

Line 311-313: When stomata close, leaves still lose water. A higher LAI in a dryland would still lead to amplified water loss, mechanistically, since there is more surface area across which the plant’s minimum conductance will shunt water to the atmosphere.

Response: Thank you for your insights. Water loss from cuticle and incompletely closed stomata is indeed important aspect. We revised this sentence as:

“The strong increases in leaf area, in turn, lead to an increase in water demand and water loss through cuticular conductance and stomatal leakiness⁴⁵, offsetting the CO₂ water saving effect due to reduced stomatal aperture^{31,42}.”

Line 333: Another good place to add “stomatal sensitivity to water deficit/drought” as another factor to consider in further/future study.

Response: We appreciate for your suggestions. This sentence now reads:

“Other factors including soil texture, rooting depth, water table depth, precipitation seasonality, and stomatal sensitivity to drought also affect ecosystem water availability and precipitation sensitivity, and their interactions with CO₂ needs to be further explored.”

References:

- Ainsworth, E.A., Long, S.P., 2005. What have we learned from 15 years of free-air CO₂ enrichment (FACE)? A meta-analytic review of the responses of photosynthesis, canopy properties and plant production to rising CO₂. *New Phytol.* 165, 351–372. <https://doi.org/10.1111/j.1469-8137.2004.01224.x>
- Harris, I., Osborn, T.J., Jones, P., Lister, D., 2020. Version 4 of the CRU TS monthly high-resolution gridded multivariate climate dataset. *Sci. Data* 7. <https://doi.org/10.1038/s41597-020-0453-3>
- Keenan, T.F., Hollinger, D.Y., Bohrer, G., Dragoni, D., Munger, J.W., Schmid, H.P., Richardson, A.D., 2013. Increase in forest water-use efficiency as atmospheric carbon dioxide concentrations rise. *Nature* 499, 324–327. <https://doi.org/10.1038/nature12291>
- Nicholson, S.E., 2011. *Dryland Climatology*, 1st ed. Cambridge University Press. <https://doi.org/10.1017/CBO9780511973840>
- Sheffield, J., Wood, E.F., Roderick, M.L., 2012. Little change in global drought over the past 60 years. *Nature* 491, 435–438. <https://doi.org/10.1038/nature11575>
- Yang, Y., Roderick, M.L., Zhang, S., McVicar, T.R., Donohue, R.J., 2019. Hydrologic implications of vegetation response to elevated CO₂ in climate projections. *Nat. Clim. Change* 9, 44–48. <https://doi.org/10.1038/s41558-018-0361-0>

Reviewer #5 (Remarks to the Author):

Summary: Zhang et al. use a combination of remotely sensed observations (NDVI and LAI estimates), global climate data, and semi-factorial model simulations of LAI (from MsTMIP) to examine the sensitivity of vegetation to precipitation, and more specifically how the sensitivity to precipitation has changed globally in both dryland and non-dryland systems. They find that the sensitivity of vegetation greenness/LAI to precipitation has increased in drylands and decreased elsewhere. They also find that these changes in vegetation sensitivity to precipitation are mostly driven by CO₂ fertilization, with increased LAI in drylands driving their enhanced sensitivity and decreased stomatal conductance in non-drylands driving their reduced sensitivity. Overall, the study seems well designed and well executed, and the authors have performed a number of sensitivity tests to ensure that their results are robust to methodological choices and assumptions. While much previous research has shown that vegetation in dry regions is more sensitive to precipitation than in wet regions, this work makes a significant and novel contribution by showing that these sensitivities have changed across much of the world (and vary by aridity) and that this is largely driven by CO₂ effects. I think the authors have mostly done a good job of responding to previous reviewer comments, and I do not have any major concerns about the work, mostly just some clarifying questions and suggestions on presentation.

Response: We appreciate the reviewer's positive comments on our manuscript.

General comments:

1) I think there's way more detail about the ecohydrological model in section 4 of the methods than is actually needed in the main text, and I would suggest moving much of it to the supplement. (Note: I don't have the relevant technical expertise to comment much on the quality/appropriateness of the ecohydrology model itself, so I'll leave that to other reviewers.) Eqn 6 seems useful to have in the main Methods section, but to me, Eqns. 7-14 seem like a little more detail than necessary for the main text. It might be easier for readers to follow and understand to have descriptions of the main logic of the model (and the sensitivities being derived from it) and save the detailed description/equations for the supplementary text, similar to what the authors did in response to the first round of reviewer comments on the DLM and MLR methods.

Response: Thank you for the suggestion. We realized that the minimalistic model used here is complex and the detailed description of the model may dilute the key message of what the model does and why it is used here. We followed the reviewer's suggestion and rewrote the section, with a brief description of the model and our major improvements to the model. The detailed description of the model is moved to the supplementary information. Please refer to the revised manuscript Line 454-479.

2) I agree with reviewer #1 (their comment on L167) that the aggregation into dryland and non-dryland at a global scale is washing out a lot of important regional variability. I understand that it's not possible in this kind of paper to delve too much into specifics of every region, but to me, it seems like a little more attention could be paid to whether and to what extent some regions deviate from the global aggregations.

Response: We appreciate the reviewer's suggestions. As we have discussed in the main text, there are other factors that may affect the precipitation sensitivity, and their effects can be strong at local scale. These effects may mask out the relatively small CO₂ effect at local scale. This is why we have to aggregate at large scale so that effects from other factors cancel with each other and the CO₂ effect pops out. In the current manuscript, we tested this at smaller scales, e.g., Fig. S10 shows the comparison between dryland and non-dryland at each continent. Drylands generally have a higher trend than the non-drylands within each continent. In this revision, we also tested the precipitation sensitivity trends within each biome (Fig. R4). 7 out of 8 biome types show a decreasing precipitation sensitivity trend along the aridity index, with the outlier being the tropical and subtropical dry broadleaf forests which only occupies less than 3% of the study area. When lumped together, the sensitivity trend – aridity index pattern is broadly consistent with the pattern we observed using all data in the study area. However, it should be noted that considerable differences still exist between biomes, which may be attributed to abiotic factors (e.g., precipitation trend, PET trend) and ecosystem characteristics. We added this figure to the supplementary information and discussed it in the main text.

“The θ_{prec} trend within each biome shows a similar pattern along the aridity index as that from the entire study regions (Supplementary Fig. S15), suggesting that biome-specific characteristics is not the major cause for the contrasting trend.”

Fig. R4. Vegetation sensitivity to precipitation along aridity index for each biome type. **a** the Olson's biome map for our study region. **b** the distribution of biome along the aridity index. **c** relationship between θ_{prec} and aridity index within each biome type.

Specific comments:

Lines 122-127: this seems to conflict with both theory (O’Gorman & Schneider 2009, Pendergrass et al. 2017) and observations (Georgi et al. 2011) of how precipitation variability changes in a warming climate. Why might this be?

Response: The precipitation variability is projected to increase as a result of increasing water holding capacity of the air as air temperature increase. This 7.5%/K scaling rate is thus the physical basis for the increase of extreme precipitation. However, this scaling rate mostly applies for each individual precipitation event, or extreme precipitation within a short period of time (e.g., hourly or daily). As the time interval increases, the scaling rate may change. In addition, many of these studies are based on model predictions, and the increasing variability is only evident over long time-scales when the signal stands out from the noises (e.g., Georgi et al. 2011). In our study, the precipitation variability is calculated based on the de-seasonalized-detrended monthly precipitation, which may yield different results than the model predictions. It should also be noted that the trend in precipitation variability is also slightly different across datasets (Fig. R5).

Fig. R5. A comparison of trend in precipitation variability between two observational based datasets. (a) precipitation from CRU TS 4.05. (b) precipitation from GPCC. (c) a scatter plot between the two datasets, x-axis is the trend from CRU and the y-axis is the trend from GPCC.

Lines 179-182: Just to clarify, are these confidence intervals accounting for both inter-model differences *and* spatial variability within the regions?

Response: For the observations, since we only have one data source, the error bar represents the 95% confidence interval of spatial variation. For modeling analysis, they represent the variations across models. These are not an apples-to-apples comparison, but considering the multiple models we have, we feel it is reasonable to present the figure this way. We have revised this sentence to clarify the meanings of error bars.

“For observations, bars represent the median trends and error bars indicate the 95% confidence interval of spatial variation through bootstrapping. For models, the bars and error bars indicate the mean and standard error of the mean (SEM) of median trend across models, respectively.”

Lines 186-194: Eqn. 1 is clearly mathematically valid, but the “well-defined ecohydrological meaning” of each term isn’t necessarily clear to me. Maybe a brief, clear explanation of those ecohydrological meanings would be helpful here?

Response: The three terms are sensitivity of LAI to transpiration ($\frac{\partial LAI}{\partial E_T}$), transpiration (E_T) sensitivity to evapotranspiration ($\frac{\partial E_T}{\partial E}$), and evapotranspiration (E) sensitivity to precipitation ($\frac{\partial E}{\partial P}$). These corresponds to the inverse of transpiration per leaf area, T:ET, and evapotranspiration ratio, respectively, but are in a partial derivative form. These terms are explained in the following paragraph (L199-L203). Here we follow the reviewer’s suggestion and briefly explain their meanings below:

“Here, θ_{prec} , approximated by LAI sensitivity to precipitation, can be further decomposed to three components with well-defined ecohydrological meaning (the inverse of transpiration per leaf area, transpiration over evapotranspiration, and evapotranspiration ratio in their partial derivative form):”

Lines 221-222: What’s the mechanism by which CO₂ could change PET?

Response: Two recent studies evaluated the change of evapotranspiration in the non-water-limited regions (e.g., wet tropical forest), and suggested that actual evapotranspiration in these regions can be regarded as the potential evapotranspiration since ET will reach PET given enough water (Milly and Dunne, 2016; Yang et al., 2019). In these vegetated wet regions, the CO₂-induced plants’ stomatal conductance decreases will increase the canopy resistance and, based on the Penman Monteith equation, lead to a reduced evapotranspiration (or PET) if other environmental factors remain unchanged. This PET considers the plants physiological response to CO₂ and it matches well with the model simulation of global runoff based on the Budyko’s framework.

Lines 234-235: The meaning of the lines in 4b is not necessarily clear from the legend in the figure. I’d suggest writing clearer descriptions of their meaning in the caption.

Response: Thank you for the suggestions, we actually made an error here. Only three CO₂ effects, instead of four, are presented here. The stepwise combinations are shown as the three lines indicated by the legend. We have revised the figure caption as below:

“Three types of CO₂ effect are considered, with each line represents a stepwise combination of them”

Lines 274-275: By “low (dry) and high (wet)”, do you mean that the CO₂ effect in low vegetation regions leads to drying of streamflow (enhanced LAI outweighs reduced conductance) and vice versa? If so, I would suggest just being a little clearer because it wasn’t immediately clear to me what “low (dry) and high (wet)” meant.

Response: Here we mean that the global runoff also shows a contrasting trend between the drylands and non-drylands. In our study area where water is the major limiting factor for vegetation, drylands have relatively low vegetation cover and non-drylands have relatively higher vegetation cover. We revised this sentence to make it clear:

“This contrasting response is also supported by the observed divergent trends in global runoff for drylands and non-drylands^{51,52}”

Lines 292-305: I like this last paragraph a lot. Great point about “greening but drying,” and a nice way to finish the paper!

Response: Thank you!

Lines 328-329: The performance of LAI3g would depend not just on NDVI3g, but also on the performance of the MODIS LAI product, correct? And in some regions the LAI estimates from MODIS can be a little suspect, I think?

Response: We agree that the LAI3g is also dependent on the performance of the MODIS LAI dataset, since the machine learning algorithm used MODIS LAI as a target during the training. There may be some regions that have less optimum LAI performance, especially in the tropics and boreal regions; however, this dataset is still one of the most widely used dataset by climate change studies (Buermann et al., 2018; Forzieri et al., 2017; Zeng et al., 2017; Zhu et al., 2016). We have revised this sentence and acknowledged the performance issues.

“Although the resulting GIMMS LAI 3g dataset strongly depends on the performance of GIMMS NDVI 3g and MODIS LAI, it compared well against field observations and partially alleviates the saturation effect of NDVI in densely vegetated regions⁶².”

Lines 408-409: Is this supposed to say “Since *not* all models participate...”? This sentence could also generally be better worded I think.

Response: We rewrote this sentence to make it clear:

“Only models that have all four simulation scenarios and predictions of LAI, E_T , and E can be used for this analysis, five models meet these requirements and are used”

Line 448: I’m not a hydrologist by any means, but it seems like interception should vary by both the amount of LAI (more LAI = more interception?) and by the intensity of precipitation (light precipitation events should have a greater percentage intercepted than heavy precipitation events, since once canopy has reached a maximum interception capacity, any additional precipitation would not be intercepted?) Is this 15% interception a common practice?

Response: Interception is indeed affected by multiple factors, including LAI, rainfall intensity, and rainfall type, simulation timestep, etc. (Wang et al., 2007) However, our current understanding of interception is still limited, and the spatial variation of interception cannot be well reproduced by the state-of-the-art land surface models. Here we used 15% as suggested by a global modeling study (Wang et al., 2007), which yielded reasonable transpiration fractions to precipitation when compared with other independent observations (Good et al., 2017). We also showed in Figure S18 that different levels of interception actually led to minor difference to our results. We discussed the potential impact in the supplementary method.

“We also tested other values and found this assumption does not directly affect our results afterwards (Supplementary Fig. S18).”

Figure S14: Would it be worth analyzing the relationship of theta-prec to VPD and/or PET trends?

Response: Thank you for your suggestion. We followed your suggestion and evaluated the relationship between θ_{prec} and VPD and PET. Considering the large differences in PET trend due to different calculation equations, we used both PET from CRU TS 4.05 (Harris et al., 2020) and PET calculated by Sheffield et al., (2012). For the VPD, we used CRU TS 4.05, which is an observational-based dataset and showed good consistency with other datasets (Yuan et al., 2019). We find very weak relationship between θ_{prec} and VPD, and θ_{prec} and PET from both datasets. The correlations are also weak within drylands and non-drylands for all three datasets. But the sign of the correlation coefficient is similar to that of temperature. This can be anticipated because both factors are directly affected by the air temperature. We replaced the original Fig. S14 with this updated Fig. R6.

Fig. R6. Relationship between trend of precipitation sensitivity and trend in climate variables (precipitation, temperature, cloud cover, potential evapotranspiration, and vapor pressure deficit). The first column shows the correlation for all pixels (**a-f**), second column for the dryland pixels (**g-l**), third column for the non-dryland pixels (**m-r**). PET from both Princeton and CRU TS 4.05 are used.

Figure S16: I really like this figure and would suggest possibly adding it as a panel in Fig. 4. Maybe instead of the soil texture analysis? To me, it seems like the soil texture analysis is more of a supplemental thing and not nearly as crucial to the main argument as Figure S16 is.

Response: Thank you for your suggestion. We did not combine these two figures since Figure S16 is based on the outputs from MsTMIP models. This Figure S16 is an extension of the Figure 3. However, Figure 4 is based on the minimalistic hydrological model. Although Figure S16 and Figure 4 are broadly consistent with each other, the minimalistic model provides a more mechanistic understanding and more details along the aridity index rather a difference just between drylands and non-drylands as shown in Figure S16. Adding Figure S16 as a subplot of Figure 4 may confuse the readers.

We agree with the reviewer that soil type is not the critical information we would like to highlight in this figure. Figure 4a aims to provide a general pattern of vegetation sensitivity to precipitation (θ_{prec}), while Figure 4b shows the trend of θ_{prec} due to direct and indirect CO₂ effect. Considering we are using a simple hydrological model and the model parameters can affect the results, we need to test whether the observed patterns are sensitivity to parameters. Soil types are important parameters in the minimalistic hydrological models and is shown here, we also tested other parameters including rooting depth and rainfall depth shown as the thin lines. The results show the observed patterns are robust and not strongly affected by the parameters.

References:

- Giorgi, F., Im, E. S., Coppola, E., Diffenbaugh, N. S., Gao, X. J., Mariotti, L., & Shi, Y. (2011). Higher hydroclimatic intensity with global warming. *Journal of Climate*, 24(20), 5309–5324.
- O’Gorman, P. A., & Schneider, T. (2009). The physical basis for increases in precipitation extremes in simulations of 21st-century climate change. *Proceedings of the National Academy of Sciences*, 106(35), 14773–14777.
- Pendergrass, A. G., Knutti, R., Lehner, F., Deser, C., & Sanderson, B. M. (2017). Precipitation variability increases in a warmer climate. *Scientific Reports*, 7, 17966.

References

- Buermann, W., Forkel, M., O’Sullivan, M., Sitch, S., Friedlingstein, P., Haverd, V., Jain, A.K., Kato, E., Kautz, M., Lienert, S., Lombardozzi, D., Nabel, J.E.M.S., Tian, H., Wiltshire, A.J., Zhu, D., Smith, W.K., Richardson, A.D., 2018. Widespread seasonal compensation effects of spring warming on northern plant productivity. *Nature* 562, 110–114.
<https://doi.org/10.1038/s41586-018-0555-7>
- Good, S.P., Moore, G.W., Miralles, D.G., 2017. A mesic maximum in biological water use demarcates biome sensitivity to aridity shifts. *Nat. Ecol. Evol.* 1, 1883.
<https://doi.org/10.1038/s41559-017-0371-8>
- Forzieri, G., Alkama, R., Miralles, D.G., Cescatti, A., 2017. Satellites reveal contrasting responses of regional climate to the widespread greening of Earth. *Science* 356, 1180–1184.
<https://doi.org/10.1126/science.aal1727>

- Milly, P.C.D., Dunne, K.A., 2016. Potential evapotranspiration and continental drying. *Nat. Clim. Change* 6, 946. <https://doi.org/10.1038/nclimate3046>
- Sheffield, J., Wood, E.F., Roderick, M.L., 2012. Little change in global drought over the past 60 years. *Nature* 491, 435–438. <https://doi.org/10.1038/nature11575>
- Wang, D., Wang, G., Anagnostou, E.N., 2007. Evaluation of canopy interception schemes in land surface models. *J. Hydrol.* 347, 308–318. <https://doi.org/10.1016/j.jhydrol.2007.09.041>
- Yuan, W., Zheng, Y., Piao, S., Ciais, P., Lombardozzi, D., Wang, Y., Ryu, Y., Chen, G., Dong, W., Hu, Z., Jain, A.K., Jiang, C., Kato, E., Li, S., Lienert, S., Liu, S., Nabel, J.E.M.S., Qin, Z., Quine, T., Sitch, S., Smith, W.K., Wang, F., Wu, C., Xiao, Z., Yang, S., 2019. Increased atmospheric vapor pressure deficit reduces global vegetation growth. *Sci. Adv.* 5, eaax1396. <https://doi.org/10.1126/sciadv.aax1396>
- Yang, Y., Roderick, M.L., Zhang, S., McVicar, T.R., Donohue, R.J., 2019. Hydrologic implications of vegetation response to elevated CO₂ in climate projections. *Nat. Clim. Change* 9, 44–48. <https://doi.org/10.1038/s41558-018-0361-0>
- Zeng, Z., Piao, S., Li, L.Z.X., Zhou, L., Ciais, P., Wang, T., Li, Y., Lian, X., Wood, E.F., Friedlingstein, P., Mao, J., Estes, L.D., Myneni, R.B., Peng, S., Shi, X., Seneviratne, S.I., Wang, Y., 2017. Climate mitigation from vegetation biophysical feedbacks during the past three decades. *Nat. Clim. Change* 7, 432–436. <https://doi.org/10.1038/nclimate3299>
- Zhu, Z., Piao, S., Myneni, R.B., Huang, M., Zeng, Z., Canadell, J.G., Ciais, P., Sitch, S., Friedlingstein, P., Arneeth, A., Cao, C., Cheng, L., Kato, E., Koven, C., Li, Y., Lian, X., Liu, Y., Liu, R., Mao, J., Pan, Y., Peng, S., Peñuelas, J., Poulter, B., Pugh, T.A.M., Stocker, B.D., Viovy, N., Wang, X., Wang, Y., Xiao, Z., Yang, H., Zaehle, S., Zeng, N., 2016. Greening of the Earth and its drivers. *Nat. Clim. Change* 6, 791–795. <https://doi.org/10.1038/nclimate3004>

Reviewer comments, third round -

REVIEWERS' COMMENTS

[Editor's note: Reviewer 4 and Reviewer 5 found the authors' responses to their comments satisfactory and had no further concerns to raise]